# An Update on Rainfall Thresholds for Rainfall-Induced Landslides in the Southern Apuan Alps (Tuscany, Italy) Using Different Statistical Methods

Roberto Giannecchini [1,2,*], Alessandro Zanon [1] and Michele Barsanti [3]

1 Department of Earth Sciences, University of Pisa, Via S. Maria 53, 56126 Pisa, Italy; alessandro.zanon94@gmail.com
2 CIRSEC—Centre for Climate Change Impact–University of Pisa, Via del Borghetto 80, 56124 Pisa, Italy
3 Department of Civil and Industrial Engineering, University of Pisa, Largo Lucio Lazzarino, 56122 Pisa, Italy; michele.barsanti@unipi.it
* Correspondence: roberto.giannecchini@unipi.it

**Abstract:** The southern Apuan Alps (Italy) are prone to rainfall-induced landslides. A first attempt to calculate rainfall thresholds was made in 2006 using non-statistical and repeatable methods for the 1975–2002 period. This research aims to update, validate, and compare the results of that attempt through different statistical approaches. Furthermore, a new dataset of rainfall and landslides from 2008 to 2016 was collected and analyzed by reconstructing the rainfall events via an automatic procedure. To obtain the rainfall thresholds in terms of the duration–intensity relationship, we applied three different statistical methods for the first time in this area: logistic regression (LR), quantile regression (QR), and least-squares linear fit (LSQ). The updated rainfall thresholds, obtained through statistical methods and related to the 1975–2002 dataset, resulted in little difference from the ones obtained with non-statistical methods and have similar efficiency values among themselves. The best one is provided by the LR, with a landslide probability of 0.55 (efficiency of 89.8%). The new rainfall thresholds, calculated by applying the three statistical methods on the dataset from 2008–2016, are similar to the 1975–2002 ones, except for the LR threshold, which exhibits a higher slope. This result confirms the validity of the thresholds obtained with the old database.

**Keywords:** rainfall threshold; shallow landslide; logistic regression; quantile regression; least-squares linear fit; southern Apuan Alps; Italy



## 1. Introduction

The need to experiment on approaches and methodologies to identify an empirical relationship between rainfall and landslides lies in the objective difficulty of identifying and quantifying the numerous factors that contribute to the initiation of landslides. If this is already difficult at slope scale, it becomes almost impossible at a basin or regional scale. On the other hand, public bodies that manage forecasting, prevention, and warning plans must often monitor large areas rather than single slopes. For this reason, the physically based approaches e.g., [1–5] are not easily applicable and are less common than the black-box approaches, in which the rainfall parameters (e.g., duration, intensity, cumulated event rainfall, antecedent rainfall) are used to find the relationships with landsliding, associating the return time of landslides to that of rainfall.

In general, the activation of landslides includes several preparatory factors (rainfall, geotechnical, hydrogeological, geological, and morphological features, as well as soil humidity, land use, etc.). Even the triggering factors may be different, however rainfall is recognized as the most common factor [6]. For these reasons, it is rare, if not impossible, to have a lot of data available for analyzing the stability conditions of slopes over a large area (e.g., using classic deterministic approaches with the factor of safety). Shallow

landslides have a strong relationship with rainfall, especially if the rainfall is intense [7–9]. Indeed, they usually involve a thin soil cover (less than 2 m thick) on steep slopes [10,11], without interference from aquifers. On the contrary, large landslides (several meters thick and covering large areas) are often related to the behaviour of water tables, whose relationship with rainfall is usually more delayed and uncertain [12]. This is the motivation for why the target of the empirical approach is superficial landslides, showing an immediate and stronger relationship with rainfall. Nevertheless, shallow landslides represent very hazardous phenomena in relation to their speed of activation, impact power, difficulty to foresee the location, and high velocity [8,13–19].

Since the well-known first attempt by [20], the calculation of critical thresholds for initiating rainfall-induced landslides became a widely used approach worldwide by using different methodologies and increasingly sophisticated statistical techniques, as shown in [6,21–42]. On the other hand, these methodologies can be reasonably suitable for predicting shallow landslide phenomena due to the general availability of rainfall data, which today are often characterized by long time series. Indeed, the good performance of these methodologies is based on the accessibility of rainfall data for many years, associated with the availability of landslide event information in terms of location and activation time.

However, the applicability of these approaches is, in our opinion, inversely proportional to the extension of the area on which they are applied. Indeed, the more the geological, morphological, hydrogeological, geotechnical, and land use features of a certain area change, the more the territory responds to rainfall change. This leads to greater uncertainty in identifying rainfall thresholds that activate superficial landslides. This specific reason suggests the necessity to apply these methods to homogeneous areas, which usually implies that the area being studied is small.

In this case, the chosen area is in the southern Apuan Alps (northwestern Tuscany, Italy). The Apuan Alps are well known not only for their beautiful environment, presence of well-known marble quarries, and proximity to popular tourist beaches (e.g., Versilia beach), but also because they are highly prone to shallow landslides and floods [14,43–45]. From this point of view, it is strategic for local authorities to be able to use warning tools based on weather forecasting and rainfall observation.

## 2. Settings, Criticisms, and Goals

The southern Apuan Alps are located in northwestern Tuscany (Italy) as seen in Figure 1. The region has one of the highest rainfall values in Italy, reaching a mean annual precipitation of about 3000 mm due to regular heavy rainstorms [44,46]. These peculiarities are mainly linked to the geographical location as well as the morphological and physiographic characteristics of the Apuan Alps. They are indeed a chain of almost 2000 m in altitude and are close and parallel to the coastline, thus intercepting the atmospheric disturbances of Atlantic origin. One of the most well-known downpours hit the southern portion of Apuan Alps on 19 June 1996, inducing hundreds of shallow landslides, debris torrents, debris flows, and floods, resulting in heavy damage to buildings, infrastructures, and the overall economy, with this incident resulting in 14 deaths (Figure 2) [14,46]. This area underwent numerous other damaging events in the past, among which the most important occurred in 1636, 1774, 1885, and 1902 [44]. In addition to the triggering cause (heavy rainstorms), there are indeed many other predisposing causes including the spreading of poorly permeable rocks (particularly schists, phyllites, and metasandstones) which lie under more permeable rocks (marbles and limestones), the spread of superficial soil cover (mainly formed by silty sand and sandy silt), the high steepness of slopes (30–45°) [14], the proximity of the mountain range to the sea (which favors low run-off times), and the abandonment of agro-forestry practices [14].

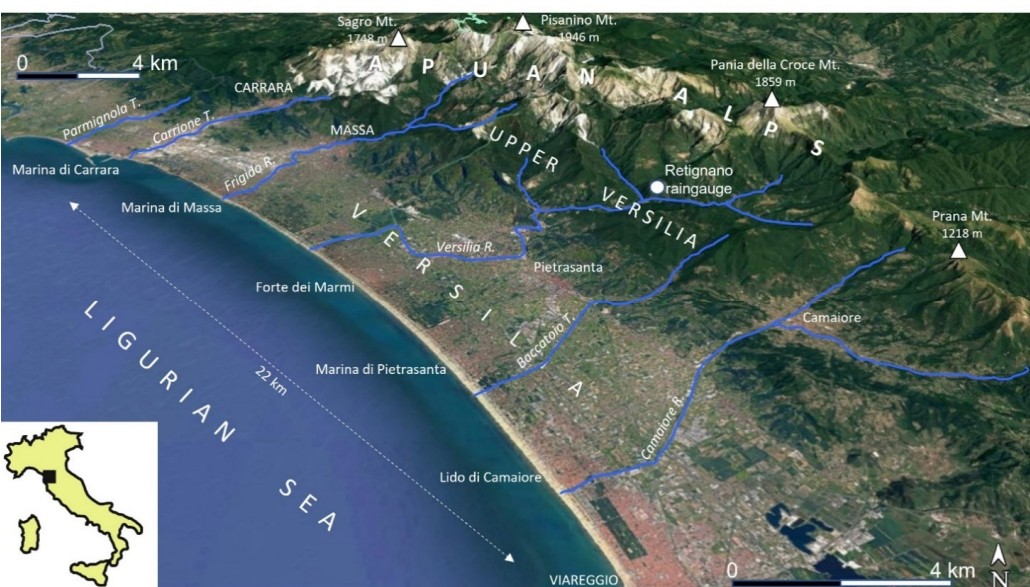

**Figure 1.** Location map and main morphological features of the Apuan Alps (white circle: Retignano raingauge; white triangles: main mountains and relative altitude in m a.s.l.; blue lines: main rivers and torrents. Being a 3D-perspective, the upper and lower scale are showed, along with the distance between two towns, as scale references (3D-perspective by Google Maps).

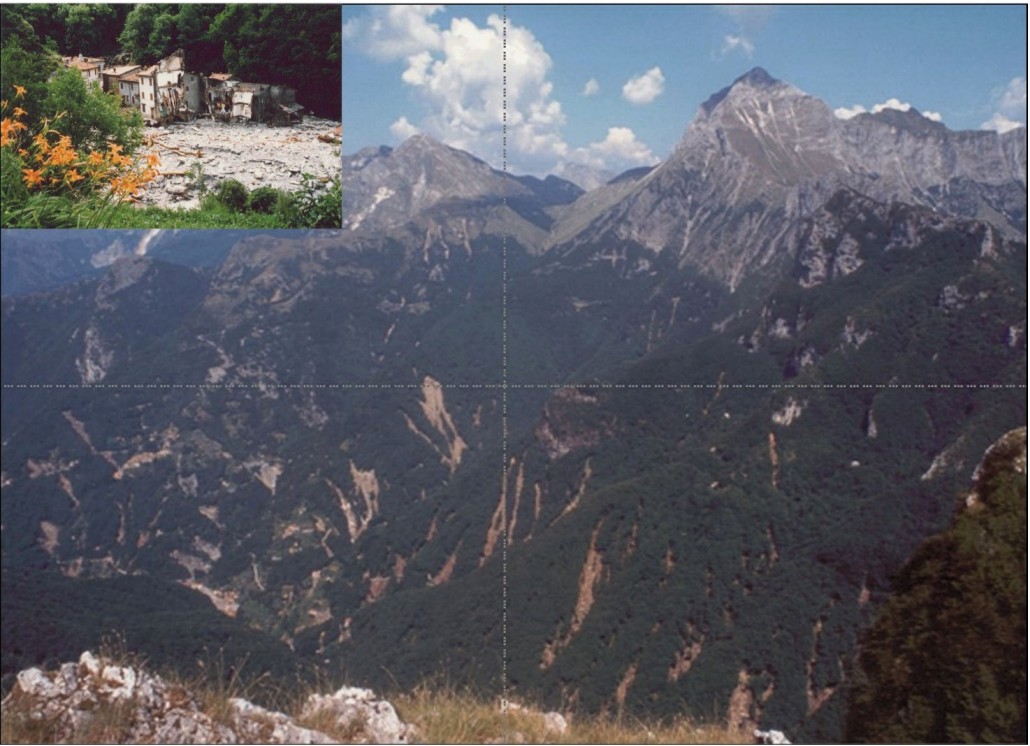

**Figure 2.** Shallow landslides activated by the rainstorm of 19 June 1996. The upper left inset shows the main village destroyed by the debris torrents, causing 13 deaths (adapted with permission from [14]. 2004, Elsevier).

The typical mass movement activated in these conditions is a very-to-extremely rapid shallow landslide, represented by an initial debris slide that evolves into a debris flow (according to the [47] classification), which then flows into the hydrographic network and brings large quantities of solid material, including tree trunks, with it. This generates a

very high destructive power when torrential debris flows interact with the anthropogenic environment, which is what occurred in the June 1996 event.

Moreover, the main feature of these events is the activation of many shallow landslides with high density (tens per square kilometers) almost simultaneously. The difficulty of predicting their location and activation time makes these phenomena very dangerous, incentivizing the scientific community to identify actions that can define alert systems. Given the lack of availability of geotechnical and hydrogeological data to define the stability conditions governing the slopes over large areas, one of the most used approaches is the quantification of rainfall quantities capable of activating shallow landslides, thus analyzing the causes rather than the effects [37,45,48–51].

With particular reference to the southern Apuan Alps, Ref. [27] already elaborated critical threshold curves with non-statistical methodologies analyzing the rainfall and shallow landslide data from 1975 to 2002 considering the raingauge of Retignano (420 m) using a non-statistical and repeatable approach, as seen in Figure 1. To update and compare the outcomes of that work, the rainfall and landslide data collected by [27] from 1975 to 2002 were analyzed using the most recent statistical approaches. Moreover, a new rainfall dataset was collected from the same raingauge over a period from 2008 to 2016 (relevant events did not occur during the 2003–2007 period, and then the research was interrupted in 2016 due to lack of resources), collecting information on shallow landslides activation, to compare the results with the 1975–2002 dataset. Therefore, the important goal of this research is the updating of the old manual rainfall thresholds with modern statistical techniques and their validation using a more recent dataset in the same area.

## 3. Materials and Methods

### 3.1. Datasets

Considering one of the aims of this research, namely the updating/comparison of the critical threshold curves of [27], it was considered fundamental to use the same raingauge (Retignano, 420 m) as the source of the rainfall data (Figure 1). At that time, the Retignano raingauge was the only one in the considered zone with a significant time series and the possibility of analyzing rainfall events lasting less than 24 h, being equipped by a pluviograph from 1975 until 1996, and only becoming electronic afterwards. On the other hand, rainstorms with durations of less than 24 h are common in the Apuan area and sometimes induce landsliding. For example, the June 1996 flood lasted just 12–13 h [14], but other events lasting only some hours activated landslides causing deaths [44].

In relation to the goals of this research, the calculation, updating, and validation of the rainfall thresholds for the area under study were carried out considering two different rainfall and landslide datasets:

- the first one is the same used by [27], namely a selection of 167 rainfall events happened during the 1975–2002 period, divided into 139 rainfall events that did not initiate shallow landslides (hereafter NSLE: no shallow landslide events) and 28 events that initiated at least one shallow landslide (hereafter SLE: shallow landslide events). These events were manually selected based on the response of the pluviographic chart, considering the start time and the end time of the rainfall for each event. For example, events with high intensity (20–30 mm/h) and low duration (1–2 h) or low intensity (2–4 mm/h) and high duration (40–50 h), as well as intermediate cases;
- the second dataset, from 2008 to 2016, was obtained by applying an automatic reconstruction of the rainfall events using the algorithm developed by [52], obtaining a total of 567 events. In order to compare this new dataset with that relating to 1975–2002, the former was further manipulated to exclude all those events of too low intensity and/or duration to be relevant for deriving the rainfall thresholds. The updated dataset resulted in 183 rainfall events, of which 152 did not activate shallow landslides (NSLE) and 31 activated shallow landslides (SLE).

For each rainfall event (both NSLE and SLE), the following information was collected: date, starting and ending time, duration (D), cumulated rainfall (E), and rainfall intensity (I = E/D).

### 3.2. Rainfall Thresholds

The most used rainfall thresholds in scientific literature, as used in this work, are based on the rainfall intensity – duration (ID) and the rainfall cumulated – duration (ED) relationships [6,28,31,37]. To obtain the rainfall ID thresholds using the two exposed datasets, three different statistical methods implemented in R software (R version 4.3.0 (21 April 2023 ucrt) were used: logistic regression (LR), quantile regression (QR), and least-squares linear fit (LSQ). Some other methods for threshold computation are found in the literature based on machine learning techniques, particularly the support vector machine (SVM) method [53] and methods based on artificial neural networks (ANN) [54,55]. We decided not to use them either because they are too complex or because a large amount of data is required to train the model. All three employed statistical methods allow for the obtaining of threshold curves with a general form of power law [20]:

$$I = \alpha \cdot D^{-\beta}$$ (1)

(I: rainfall intensity (mm/h); D: duration (h), which are straight lines in log-log axis; $\alpha$: intercept on the ordinate axis corresponding to 1 h; $\beta$: threshold curve slope).

LR models (see also [45]) are a particular form of generalized linear models in which the value of the output continuous variable, given the set of predictors (in this case duration and intensity), can vary between 0 and 1. It is interpreted as probability p that a rainfall of duration D and intensity I can trigger a landslide. This is definable by the equation:

$$p = \frac{1}{1 + \exp[-(\beta_0 + \beta_1 D' + \beta_2 I')]}$$ (2)

(I': $Log_{10}$ (I) (mm/h); D': $Log_{10}$(D) (h); $\beta i$ with i = 0, 1, 2: parameters of the regression coefficients).

The determination of the rainfall thresholds with LR requires the use of both types of rainfall events, attributing *a posteriori* probability p = 1 to the SLE events and p = 0 to the NSLE events.

QR and LSQ were used by several authors to determine ID and ED thresholds [31,34,37,56,57] considering only the SLE events as input data. In the LSQ method, a linear dependence of type I' = $\alpha$ + $\beta$D' is assumed and the calculation of $\alpha$ and $\beta$ parameters is carried out by minimizing the sum of the square deviations between the measured I' and the fit-forecasted value, keeping the detected D' values fixed. In this way, it is possible to determine the threshold TLSQ-50 corresponding to the fiftieth Gaussian percentile of the SLE events. Moreover, the standard deviation ($\sigma$) is obtained, and it is therefore possible to determine the lower percentiles by translating the intercept $\alpha$ using $\sigma$. To obtain several thresholds, we define a probability $p_e$ (called exceedance probability threshold [34]) corresponding to a straight line below which, on average, $p_e \cdot N$ events are found, where N is the number of SLE events. The $\alpha$ value corresponding to the chosen $p_e$ is calculated, translating the TLSQ_50 line according to the procedure described in [56].

Since the LSQ method allows for the calculation of different thresholds with different $\alpha$ values keeping $\beta$ unchanged, it properly represents the general trend of the distribution of events, but, in some cases, does not obtain good thresholds for low $p_e$ values. For this reason, the QR method was used as well. Once $p_e$ is chosen, the QR method calculates the $\alpha$ and the $\beta$ of the threshold below which, on average, $p_e \cdot N$ events are found. QR is considered a robust technique and a valid alternative to the LSQ especially when outliers are present. QR also uses only the SLE events as input data, but unlike LR and LSQ methods, in QR both the $\alpha$ and the $\beta$ values depend on the $p_e$ value. For this reason, the obtained thresholds exhibit different slopes.

Since the methods used for determining the thresholds are probabilistic and not deterministic, it is necessary to carry out a statistical check. This validation allows us to evaluate the possibility of their concrete use in an alert system and to get a choice criterion for an optimal threshold.

The validation of the rainfall thresholds was pursued using the approach suggested by [58], using the number of TP (true positive), TN (true negative), FP (false positive), and FN (false negative) to calculate the following skill scores:

- POD (probability of detection): TP/(TP + FN) [range 0, 1—optimal value 1];
- POFD (probability of false detection): FP/(FP + TN) [range 0, 1—optimal value 0];
- POFA (probability of false alarm): FP/(TP + FP) [range 0, 1—optimal value 0];
- Ef (efficiency): (TP + TN)/(FP + FN + TP + TN) [range 0, 1—optimal value 1];
- HK ([59] skill score): [TP/(TP + FN)]–[FP/(FP + TN)] or POD-POFD [range −1, 1—optimal value 1].

## 4. Results

### 4.1. Processing of the 1975–2002 Dataset

As explained previously, the 1975–2002 dataset includes 167 rainfall events, 28 of which induced shallow landslides. The main rainstorms in terms of landslides number occurred in 1984, 1992, 1994, 1996, 1998, and 2000. An analysis of the event distribution in relation to seasons show that they occurred in autumn 40% of the time [27].

Using the 1975–2002 rainfall and landslide dataset [27], the rainfall thresholds for the southern Apuan Alps were recalculated using the three fit methods (LR, QR, LSQ) by means of R software [60]. The values of $\alpha$ and $\beta$ were obtained using the three methods along with their 95% confidence intervals, and are reported in Tables 1–3. The processing of the rainfall and landslide dataset produced the contingency and skill score tables shown in Tables 4–6 and Figures 3–5. The best results of the three statistical methods and the lower threshold of [27] are reported in Table 7. The graph shown in Figure 6 allows for a visual comparison between the best thresholds obtainable with the three aforementioned methods and the two thresholds plotted by [27].

**Table 1.** Values of $\alpha$ and $\beta$ for the 1975–2002 rainfall and landslide dataset obtained with the LR method for different landslide probability p (C.I.: confidence interval).

| p | $\alpha$ | 95% C.I. for $\alpha$ | $\beta$ | 95% C.I. for $\beta$ |
|---|---|---|---|---|
| 0.01 | 16.0 | [8.63, 26.5] | −0.60 | [−0.70, −0.46] |
| 0.05 | 23.0 | [14.9, 33.2] | −0.60 | [−0.70, −0.46] |
| 0.10 | 27.1 | [18.9, 37.3] | −0.60 | [−0.70, −0.46] |
| 0.15 | 30.0 | [21.6, 40.1] | −0.60 | [−0.70, −0.46] |
| 0.20 | 32.4 | [24.2, 42.6] | −0.60 | [−0.70, −0.46] |
| 0.50 | 44.0 | [34.7, 55.1] | −0.60 | [−0.70, −0.46] |

**Table 2.** Values of $\alpha$ and $\beta$ for the 1975–2002 rainfall and landslide dataset obtained with the QR method for different exceeding probability $p_e$ (C.I.: confidence interval).

| $p_e$ | $\alpha$ | 95% C.I. for $\alpha$ | $\beta$ | 95% C.I. for $\beta$ |
|---|---|---|---|---|
| 0.01 | 9.56 | [2.80, 32.6] | −0.33 | [−0.75, +0.085] |
| 0.05 | 28.4 | [8.36, 96.7] | −0.65 | [−1.0, −0.25] |
| 0.10 | 27.9 | [10.2, 76.7] | −0.60 | [−0.94, −0.27] |
| 0.15 | 27.6 | [14.4, 53.0] | −0.58 | [−0.80, −0.35] |
| 0.20 | 29.3 | [19.3, 44.2] | −0.60 | [−0.77, −0.43] |
| 0.50 | 52.0 | [32.8, 82.4] | −0.75 | [−0.94, −0.56] |

**Table 3.** Values of α and β for the 1975–2002 rainfall and landslide dataset obtained with the LSQ method for different exceeding probability $p_e$ (C.I.: confidence interval).

| $p_e$ | α | 95% C.I. for α | β | 95% C.I. for β |
|---|---|---|---|---|
| 0.01 | 17.4 | [11.8, 25.6] | −0.69 | [−0.87, −0.51] |
| 0.05 | 23.5 | [16.0, 34.7] | −0.69 | [−0.87, −0.51] |
| 0.10 | 27.7 | [18.8, 40.8] | −0.69 | [−0.87, −0.51] |
| 0.15 | 30.8 | [20.9, 45.4] | −0.69 | [−0.87, −0.51] |
| 0.20 | 33.6 | [22.8, 49.6] | −0.69 | [−0.87, −0.51] |
| 0.50 | 48.9 | [33.2, 72.0] | −0.69 | [−0.87, −0.51] |

**Table 4.** Contingency table (TP, FN, FP, TN) and skill scores (POD, POFD, POFA, Ef, HK) of the thresholds obtained by LR method. Threshold showing the maximum efficiency value is highlighted in red.

| Threshold | p | TP | FN | FP | TN | POD | POFD | POFA | Ef | HK |
|---|---|---|---|---|---|---|---|---|---|---|
| TLR-5 | 0.05 | 27 | 1 | 81 | 58 | 0.964 | 0.583 | 0.750 | 0.509 | 0.382 |
| TLR-10 | 0.10 | 26 | 2 | 45 | 94 | 0.929 | 0.324 | 0.634 | 0.719 | 0.605 |
| TLR-30 | 0.30 | 16 | 12 | 19 | 120 | 0.571 | 0.137 | 0.543 | 0.814 | 0.435 |
| TLR-50 | 0.50 | 12 | 16 | 2 | 137 | 0.429 | 0.014 | 0.143 | 0.892 | 0.414 |
| TLR-55 | 0.55 | 12 | 16 | 1 | 138 | 0.429 | 0.007 | 0.077 | 0.898 | 0.421 |
| TLR-70 | 0.70 | 5 | 23 | 1 | 138 | 0.179 | 0.007 | 0.167 | 0.856 | 0.171 |
| TLR-90 | 0.90 | 2 | 26 | 0 | 139 | 0.071 | 0.000 | 0.000 | 0.844 | 0.071 |

**Table 5.** Contingency table (TP, FN, FP, TN) and skill scores (POD, POFD, POFA, Ef, HK) of the thresholds obtained using QR method. Threshold showing the maximum efficiency value is highlighted in red (TQR_60 shows the same results but is calculated for a higher probability).

| Threshold | $p_e$ | TP | FN | FP | TN | POD | POFD | POFA | Ef | HK |
|---|---|---|---|---|---|---|---|---|---|---|
| TQR_01 | 0.01 | 26 | 2 | 107 | 32 | 0.929 | 0.770 | 0.805 | 0.347 | 0.159 |
| TQR_05 | 0.05 | 27 | 1 | 57 | 82 | 0.964 | 0.410 | 0.679 | 0.653 | 0.554 |
| TQR_10 | 0.10 | 25 | 3 | 45 | 94 | 0.893 | 0.324 | 0.643 | 0.713 | 0.569 |
| TQR_15 | 0.15 | 25 | 3 | 36 | 103 | 0.893 | 0.259 | 0.590 | 0.766 | 0.634 |
| TQR_20 | 0.20 | 24 | 4 | 33 | 106 | 0.857 | 0.237 | 0.579 | 0.778 | 0.620 |
| TQR_50 | 0.50 | 13 | 14 | 13 | 126 | 0.481 | 0.094 | 0.500 | 0.837 | 0.388 |
| TQR_55 | 0.55 | 12 | 15 | 4 | 136 | 0.444 | 0.029 | 0.250 | 0.886 | 0.416 |
| TQR_60 | 0.60 | 12 | 15 | 4 | 136 | 0.444 | 0.029 | 0.250 | 0.886 | 0.416 |
| TQR_65 | 0.65 | 11 | 17 | 3 | 136 | 0.393 | 0.022 | 0.214 | 0.880 | 0.371 |
| TQR_90 | 0.90 | 2 | 26 | 0 | 139 | 0.071 | 0.000 | 0.000 | 0.844 | 0.071 |

**Table 6.** Contingency table (TP, FN, FP, TN) and skill scores (POD, POFD, POFA, Ef, HK) of the thresholds obtained using LSQ method. Threshold showing the maximum efficiency value is highlighted in red.

| Threshold | $p_e$ | TP | FN | FP | TN | POD | POFD | POFA | Ef | HK |
|---|---|---|---|---|---|---|---|---|---|---|
| TLSQ_01 | 0.01 | 27 | 1 | 137 | 2 | 0.964 | 0.986 | 0.835 | 0.174 | −0.021 |
| TLSQ_05 | 0.05 | 27 | 1 | 115 | 24 | 0.964 | 0.827 | 0.810 | 0.305 | 0.137 |
| TLSQ_10 | 0.10 | 27 | 1 | 88 | 51 | 0.964 | 0.633 | 0.765 | 0.467 | 0.331 |
| TLSQ_15 | 0.15 | 26 | 2 | 67 | 72 | 0.929 | 0.482 | 0.720 | 0.587 | 0.447 |
| TLSQ_20 | 0.20 | 24 | 4 | 47 | 92 | 0.857 | 0.338 | 0.662 | 0.695 | 0.519 |
| TLSQ_50 | 0.50 | 14 | 14 | 9 | 130 | 0.500 | 0.065 | 0.391 | 0.862 | 0.435 |
| TLSQ_55 | 0.55 | 14 | 14 | 4 | 135 | 0.500 | 0.029 | 0.222 | 0.892 | 0.471 |
| TLSQ_60 | 0.60 | 11 | 17 | 3 | 136 | 0.393 | 0.022 | 0.214 | 0.880 | 0.371 |
| TLSQ_65 | 0.65 | 8 | 20 | 2 | 137 | 0.286 | 0.014 | 0.200 | 0.868 | 0.271 |
| TLSQ_90 | 0.90 | 2 | 26 | 0 | 139 | 0.071 | 0.000 | 0.000 | 0.844 | 0.071 |

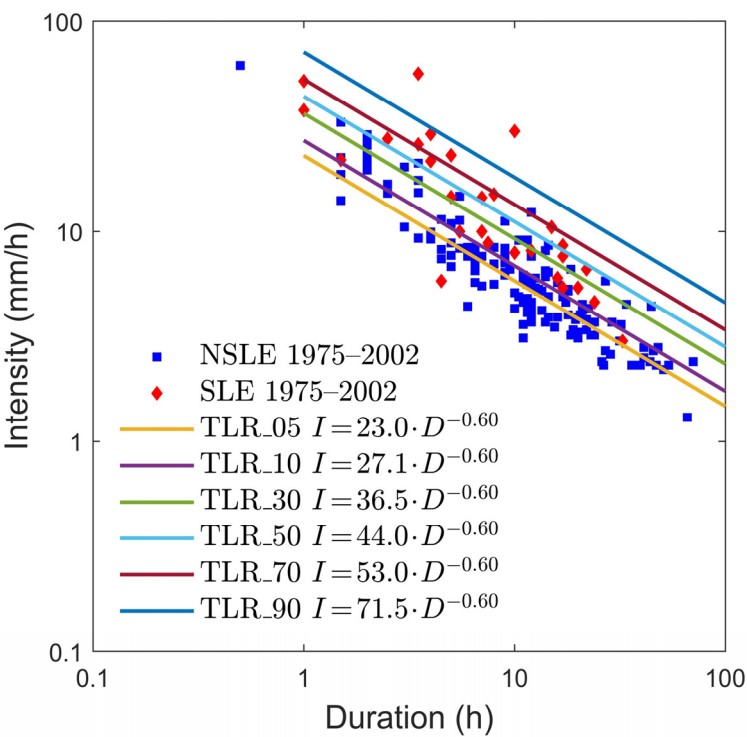

**Figure 3.** Rainfall thresholds obtained using LR method for the 1975–2002 period and corresponding to different landslide probability (5–90%). Blue squares: events that did not trigger landslides (NSLE); red lozenges: events that triggered landslides (SLE).

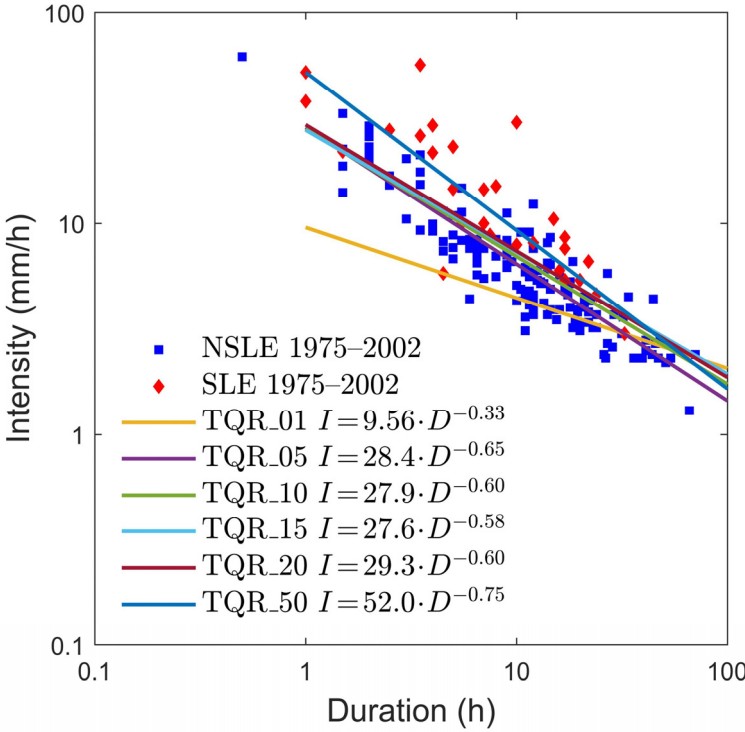

**Figure 4.** Rainfall thresholds obtained using QR method for the 1975–2002 period and corresponding to different landslide probability (1–50%). Blue squares: events that did not trigger landslides (NSLE); red lozenges: events that triggered landslides (SLE).

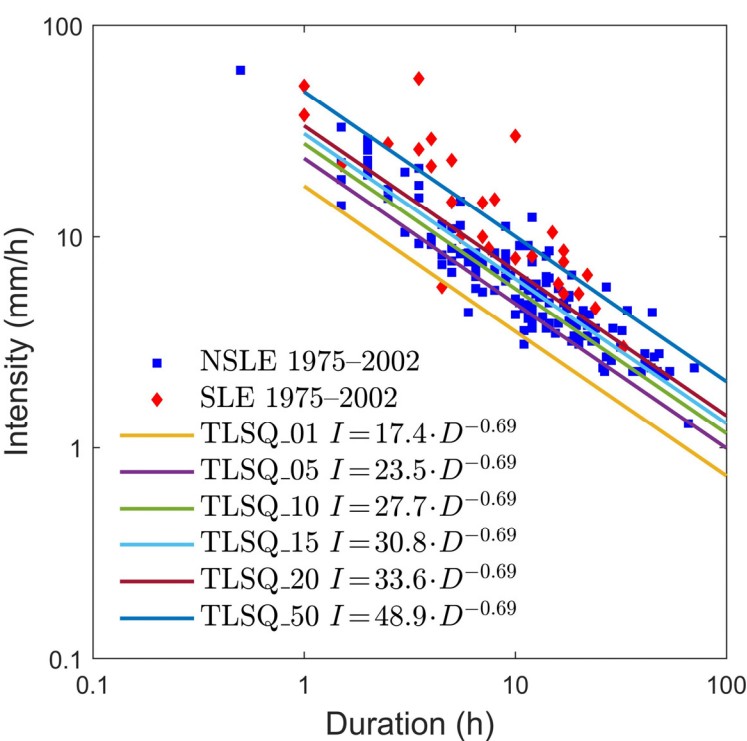

**Figure 5.** Rainfall thresholds obtained using the LSQ method for the 1975–2002 period and corresponding to different landslide probabilities (1–50%). Blue squares: events that did not trigger landslides (NSLE); red lozenges: events that triggered landslides (SLE).

**Table 7.** Summary of the contingencies (TP, FN, FP, TN) and skill scores (POD, POFD, POFA, Ef, HK) of the SLE and NSLE forecasts obtained from the three best thresholds calculated on the 1975–2002 dataset using LR, LSQ, QR statistical techniques and the lower rainfall threshold plotted by [27] (blue in Figure 6). In this case, both the number of correct forecasts (28 out of 28 for SLE) and the number of false alarms (FP) are particularly high.

| Threshold | $p, p_e$ | TP | FN | FP | TN | POD | POFD | POFA | Ef | HK |
|---|---|---|---|---|---|---|---|---|---|---|
| Low T 2006 | - | 28 | 0 | 56 | 83 | 1.000 | 0.513 | 0.666 | 0.664 | 0.606 |
| TLR_55 | 0.55 | 12 | 16 | 1 | 138 | 0.429 | 0.007 | 0.077 | 0.898 | 0.421 |
| TQR_55 | 0.55 | 12 | 15 | 4 | 136 | 0.444 | 0.029 | 0.250 | 0.886 | 0.416 |
| TLSQ_55 | 0.55 | 14 | 14 | 4 | 135 | 0.500 | 0.029 | 0.222 | 0.892 | 0.471 |

The comparison between the thresholds obtained by [27] using a manual fitting and those obtained in this study by statistical calculation on the same rainfall and landslide dataset (1975–2002) was carried out by evaluating the efficiency of the lower threshold only of [27] (straight blue line in Figure 6). The upper threshold (red line in Figure 6) was not considered in the comparison due to the scarcity of rainstorms that induced many landslides (type A events according to [27]). From the position of the lower threshold and joining the events of type A and type B (rainstorms that induced a few landslides) events in a single category of landslide events (SLE), the contingency and skill scores table were calculated (Table 7). Figure 6 shows the comparison between the thresholds of [27] and those obtained using statistical methodologies. It is evident the very low position of the low threshold (blue line) of such Author, which, on the other hand, was considered conservative. Indeed, many NSLE events (blue squares) fall above the threshold. The implication and the consequence of its low position is highlighted in Table 7, showing the total accuracy of SLE's prediction while also showing the fact that it generates a notable number of false alarms (56 events). This characteristic clearly affects the efficiency (Ef) of the threshold, which is around 66%.

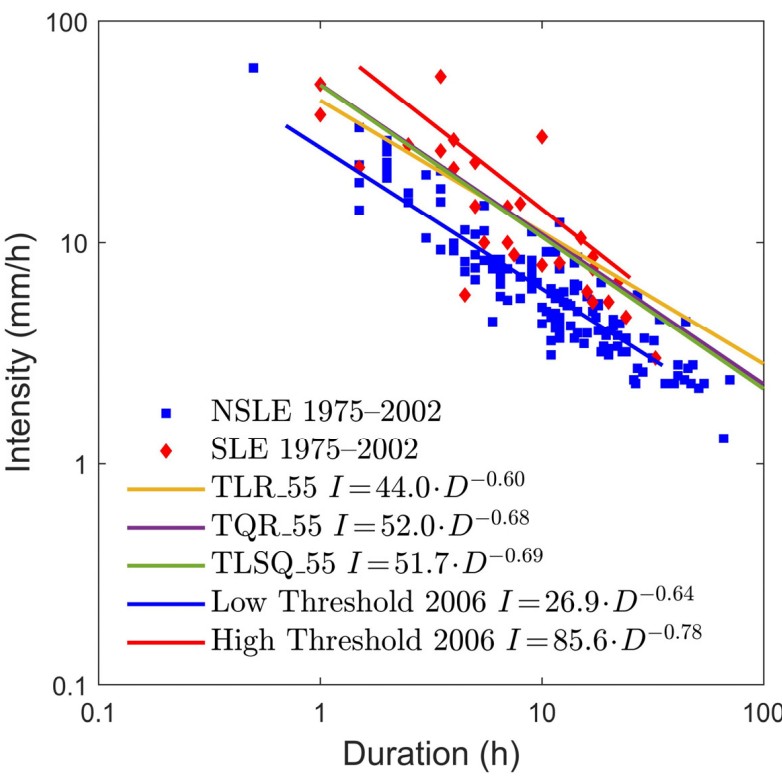

**Figure 6.** Comparison among the upper and lower rainfall thresholds of [27] (red and blue lines, respectively) and those obtained by the three statistical methods (yellow, green, violet lines) considering the maximum efficiency related to 1975–2002 period. Blue squares: events that did not trigger landslides (NSLE); red lozenges: events that triggered landslides (SLE).

The rainfall thresholds obtained through statistical methods, being higher in the analyzed dataset, determine contingency values very differently from the lower threshold of [27]. The results (Table 7) show that the three thresholds have very similar efficiency values, differing from each other by around 0.1%. The best of the three thresholds is that obtained through the LR method, with a landslide probability of 0.55. It guarantees an efficiency of 89.8%, with a correct forecast of 150 events out of a total of 167.

### 4.2. Processing of the 2008–2016 Dataset

Through the application of the algorithm determined in [52], the automatic reconstruction of the rainfall events was allowed to collect 567 main rainstorms that occurred in the southern Apuan Alps from 2008 to 2016. The events show the following distribution: 21% occurred in winter (December, January, February); 30% in spring (March, April, May); 22% in summer (June, July, August); and 27% in autumn (September, October, November) (Figure 7).

An archive search was carried out for the 567 rainfall events in order to identify the number of shallow landslides triggered, with a total of 93 being found. Many of them were triggered during the same rainstorm, such as the 9 failures activated by the 25 October 2011 thunderstorm or the 11 failures triggered during the 5 January 2014 event. Therefore, a total of 31 rainfall events which induced at least one shallow landslide (SLE) were obtained.

The seasonal distribution of the SLEs in the 2008–2016 period follows what was already observed in [27] during the 1975–2002 period, with a higher frequency in autumn (from September to November, 42%). However, a high landslide hazard of 39% was also recognized in winter (December-January-February). In spring and summer, the number of SLE decreases to values of 18% and 1%, respectively (Figure 8).

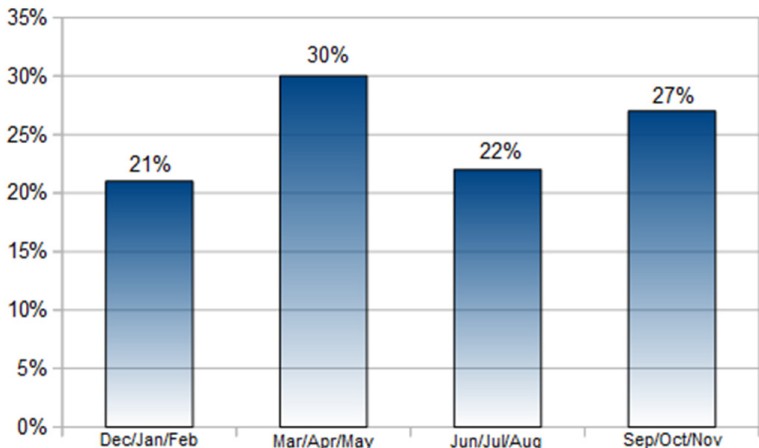

**Figure 7.** Distribution of main rainfall events that occurred in the southern Apuan Alps from 2008 to 2016 in relation to season.

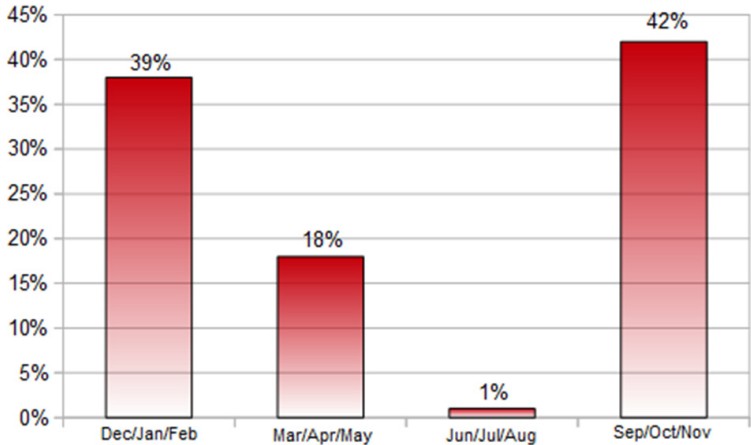

**Figure 8.** Distribution of rainfall events inducing shallow landslides (SLE) in the southern Apuan Alps from 2008 to 2016 in relation to season.

Figure 9 shows all 567 events automatically reconstructed by the algorithm, including many events characterized by low values of I and D (blue events under the discriminating green line, obtained according to the method explained in [37]). Using the whole set of 567 rainfall events within the 2008–2016 period, the values of the efficiency of the thresholds are biased. Computing the thresholds efficiencies, it is noted that the value of approximately 95% for the best one is due to the high number of true negative events (TN). To get around the problem of the presence of rainfall events with too low of an intensity and duration, making them insignificant for the calculation of the rainfall thresholds, these events were removed from the dataset. Only SLEs and NSLEs above the discriminating line of the equation I = 8.173 × D$^{-0.547}$ (green line in Figure 9) were considered valid for the computation of skill scores and contingency tables. This choice discards the events with an intensity of less than 10 mm/h in 1 h for short duration events and 1 mm/h rainfall in 100 h for long-duration events. In this way, the original data set consisting of 567 events (relating to the 2008–2016 period) was reduced to 183 events (Figure 10).

As for the first dataset (1975–2002), the three fit methods (LR, QR, LSQ) were applied even to the 2008–2016 dataset, computing the thresholds using the R software (Figures 11–13). The values of α and β obtained using the three methods, together with their 95% confidence intervals, are reported in Tables 8–10.

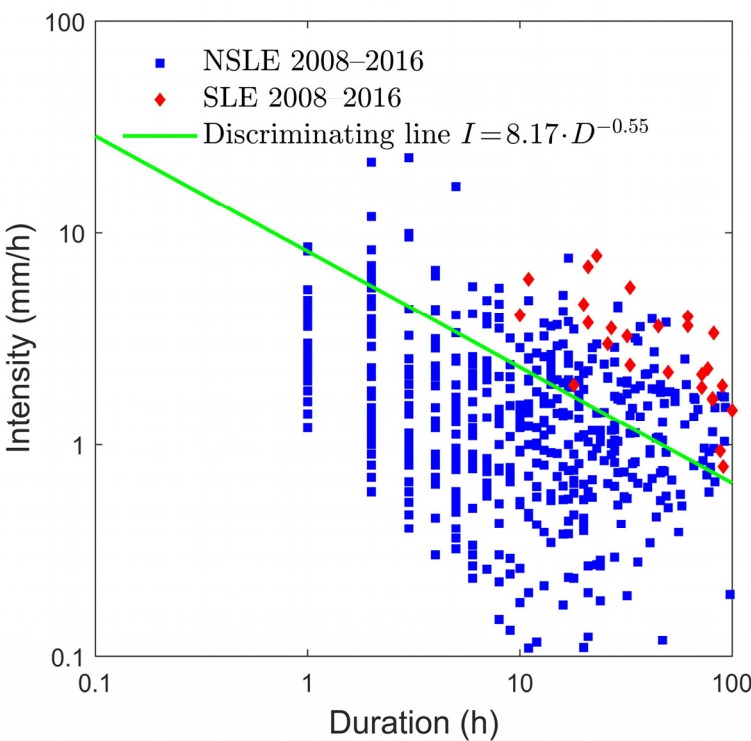

**Figure 9.** ID log-log plot of rainfall events SLEs and NSLEs of the 2008–2016 period reconstructed using the algorithm [52]. Green line separates events considered significant (above the green line) from events that instead cause a bias in the results (below the green line) due to an increase of the number of TN in the contingency tables and, therefore, a considerable increase of the efficiency values (Ef). Blue squares: events that did not trigger landslides (NSLE); red lozenges: events that triggered landslides (SLE).

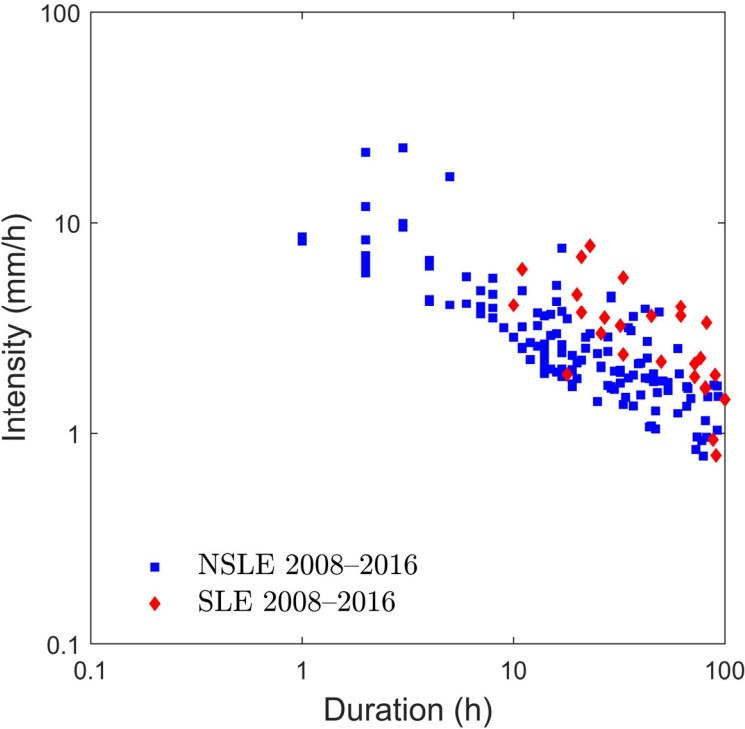

**Figure 10.** ID log-log plot of rainfall events SLEs and NSLEs of the 2008–2016 period located above the discriminating line shown in Figure 9. Blue squares: events that did not trigger landslides (NSLE); red lozenges: events that triggered landslides (SLE).

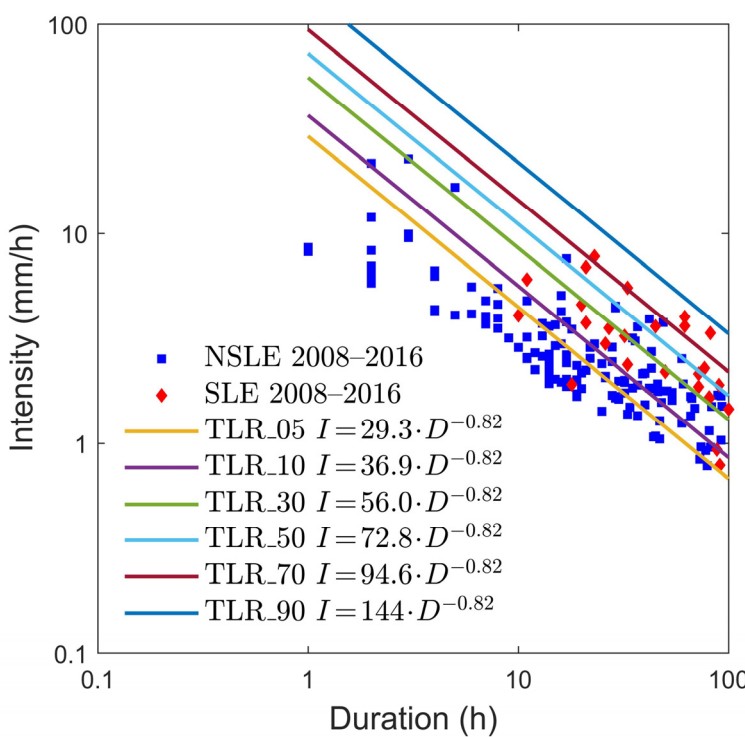

**Figure 11.** Rainfall thresholds obtained by using the LR method for the 2008–2016 period and corresponding to different landslide probabilities (5–90%). Blue squares: events that did not trigger landslides (NSLE); red lozenges: events that triggered landslides (SLE).

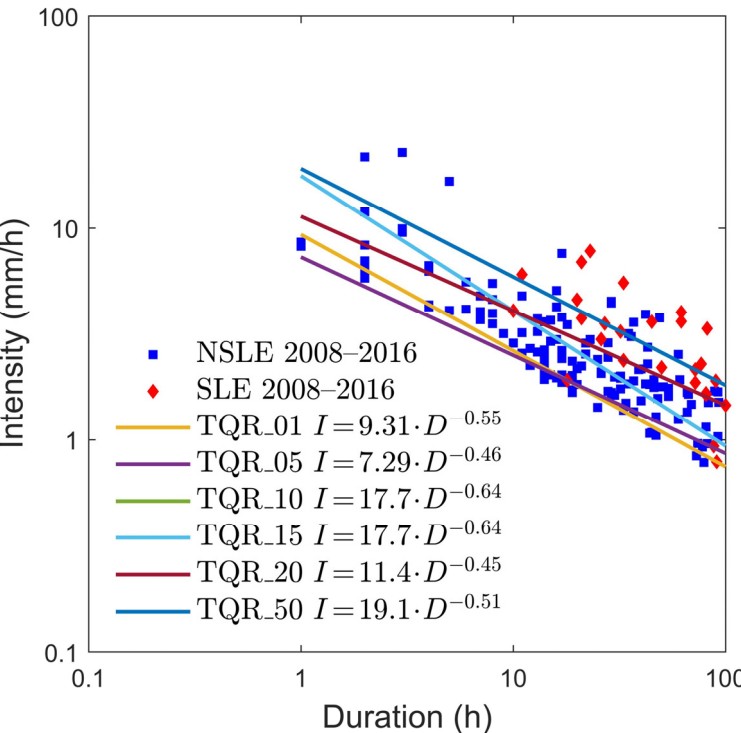

**Figure 12.** Rainfall thresholds obtained using the QR method for the 2008–2016 period and corresponding to different landslide probabilities (1–50%). The TQR_10 and TQR_15 lines are perfectly superimposed (only the light blue one is shown). Blue squares: events that did not trigger landslides (NSLE); red lozenges: events that triggered landslides (SLE).

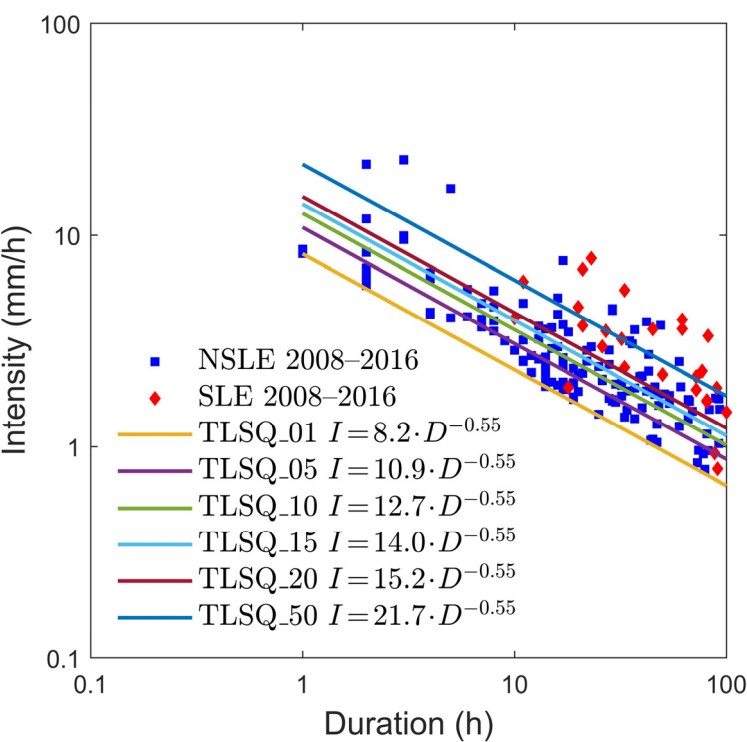

**Figure 13.** Rainfall thresholds obtained using the LSQ method for the 2008–2016 period and corresponding to different landslide probabilities (1–50%). Blue squares: events that did not triggered landslides (NSLE); red lozenges: events that triggered landslides (SLE).

**Table 8.** Values of $\alpha$ and $\beta$ for the 2008–2016 rainfall and landslide dataset obtained with the LR method for different landslide probability p (C.I.: confidence interval).

| p | $\alpha$ | 95% C.I. for $\alpha$ | $\beta$ | 95% C.I. for $\beta$ |
|---|---|---|---|---|
| 0.01 | 17.6 | [9.73, 42.1] | −0.82 | [−1.1, −0.66] |
| 0.05 | 29.3 | [17.2, 63.9] | −0.82 | [−1.1, −0.66] |
| 0.10 | 36.9 | [21.2, 79.2] | −0.82 | [−1.1, −0.66] |
| 0.15 | 42.5 | [23.9, 99.7] | −0.82 | [−1.1, −0.66] |
| 0.20 | 47.4 | [26.5, 119] | −0.82 | [−1.1, −0.66] |
| 0.50 | 72.8 | [38.8, 193] | −0.82 | [−1.1, −0.66] |

**Table 9.** Values of $\alpha$ and $\beta$ for the 2008–2016 rainfall and landslide dataset obtained with the QR method for different exceeding probability $p_e$ (C.I.: confidence interval).

| $p_e$ | $\alpha$ | 95% C.I. for $\alpha$ | $\beta$ | 95% C.I. for $\beta$ |
|---|---|---|---|---|
| 0.01 | 9.31 | [1.84, 47.0] | −0.55 | [−0.93, −0.17] |
| 0.05 | 7.29 | [1.72, 30.9] | −0.46 | [−0.81, −0.12] |
| 0.10 | 17.7 | [3.40, 91.7] | −0.64 | [−1.1, −0.23] |
| 0.15 | 17.7 | [3.67, 84.9] | −0.64 | [−1.1, −0.23] |
| 0.20 | 11.4 | [2.73, 47.1] | −0.45 | [−0.81, −0.079] |
| 0.50 | 19.1 | [8.37, 43.5] | −0.51 | [−0.71, −0.31] |

Tables 11–13 show the contingency tables and skill scores obtained. The most efficient thresholds are reported in Table 14, in which the efficiencies of the thresholds before and after the data selection using the discriminating line are compared. The lines shown in Figure 14 allows the visual comparison of the three thresholds with the best efficiency.

**Table 10.** Values of α and β for the 2008–2016 rainfall and landslide dataset obtained with the LSQ method for different exceeding probability $p_e$ (C.I.: confidence interval).

| $p_e$ | α | 95% C.I. for α | β | 95% C.I. for β |
|---|---|---|---|---|
| 0.01 | 8.17 | [3.86, 17.3] | −0.55 | [−0.73, −0.36] |
| 0.05 | 10.9 | [5.14, 23.0] | −0.55 | [−0.73, −0.36] |
| 0.10 | 12.7 | [5.98, 26.8] | −0.55 | [−0.73, −0.36] |
| 0.15 | 14.0 | [6.63, 29.7] | −0.55 | [−0.73, −0.36] |
| 0.20 | 15.2 | [7.20, 32.3] | −0.55 | [−0.73, −0.36] |
| 0.50 | 21.7 | [10.2, 45.9] | −0.55 | [−0.73, −0.36] |

**Table 11.** Contingency table (TP, FN, FP, TN) and skill scores (POD, POFD, POFA Ef, HK) for the thresholds obtained by the LR method. The threshold showing the maximum efficiency (Ef = 0.885) is highlighted in red. NA: not available.

| Threshold | p | TP | FN | FP | TN | POD | POFD | POFA | Ef | HK |
|---|---|---|---|---|---|---|---|---|---|---|
| TLR_5 | 0.05 | 29 | 2 | 81 | 71 | 0.935 | 0.533 | 0.736 | 0.546 | 0.403 |
| TLR_10 | 0.10 | 27 | 4 | 57 | 95 | 0.871 | 0.375 | 0.679 | 0.667 | 0.496 |
| TLR_30 | 0.30 | 18 | 13 | 19 | 133 | 0.581 | 0.125 | 0.514 | 0.825 | 0.456 |
| TLR_50 | 0.50 | 13 | 18 | 3 | 149 | 0.419 | 0.020 | 0.188 | 0.885 | 0.400 |
| TLR_70 | 0.70 | 7 | 24 | 0 | 152 | 0.226 | 0.000 | 0.000 | 0.869 | 0.226 |
| TLR_90 | 0.90 | 0 | 31 | 0 | 152 | 0.000 | 0.000 | NA | 0.831 | 0.000 |

**Table 12.** Contingency table (TP, FN, FP, TN) and skill scores (POD, POFD, POFA Ef, HK) for the thresholds obtained using the QR method. Although the threshold relating to the ninetieth percentile of exceedance probability shows the highest efficiency (Ef = 0.852), the threshold highlighted in red was considered better for the higher POD and HK parameters.

| Threshold | $p_e$ | TP | FN | FP | TN | POD | POFD | POFA | Ef | HK |
|---|---|---|---|---|---|---|---|---|---|---|
| TQR_01 | 0.01 | 31 | 0 | 130 | 22 | 1.000 | 0.855 | 0.807 | 0.290 | 0.145 |
| TQR_05 | 0.05 | 28 | 3 | 128 | 24 | 0.903 | 0.842 | 0.821 | 0.284 | 0.061 |
| TQR_10 | 0.10 | 28 | 3 | 69 | 83 | 0.903 | 0.454 | 0.711 | 0.607 | 0.449 |
| TQR_15 | 0.15 | 28 | 3 | 69 | 83 | 0.903 | 0.454 | 0.711 | 0.607 | 0.449 |
| TQR_20 | 0.20 | 25 | 5 | 38 | 114 | 0.833 | 0.250 | 0.603 | 0.764 | 0.583 |
| TQR_50 | 0.50 | 15 | 16 | 13 | 139 | 0.484 | 0.086 | 0.464 | 0.842 | 0.398 |
| TQR_55 | 0.55 | 13 | 18 | 12 | 140 | 0.419 | 0.079 | 0.480 | 0.836 | 0.340 |
| TQR_60 | 0.60 | 13 | 18 | 11 | 141 | 0.419 | 0.072 | 0.458 | 0.842 | 0.347 |
| TQR_65 | 0.65 | 10 | 21 | 10 | 142 | 0.323 | 0.066 | 0.500 | 0.831 | 0.257 |
| TQR_90 | 0.90 | 4 | 27 | 0 | 152 | 0.129 | 0.000 | 0.000 | 0.852 | 0.129 |

**Table 13.** Contingency table (TP, FN, FP, TN) and skill scores (POD, POFD, POFA, Ef, HK) of the thresholds obtained using the LSQ method. Maximum efficiency value is reached with the threshold highlighted in red relating to an exceedance probability of 0.45 (Ef = 0.852).

| Threshold | $p_e$ | TP | FN | FP | TN | POD | POFD | POFA | Ef | HK |
|---|---|---|---|---|---|---|---|---|---|---|
| TLSQ_01 | 0.01 | 31 | 0 | 152 | 0 | 1.000 | 1.000 | 0.831 | 0.169 | 0.000 |
| TLSQ_05 | 0.05 | 28 | 3 | 97 | 55 | 0.903 | 0.638 | 0.776 | 0.454 | 0.265 |
| TLSQ_10 | 0.10 | 27 | 4 | 75 | 77 | 0.871 | 0.493 | 0.735 | 0.568 | 0.378 |
| TLSQ_15 | 0.15 | 27 | 4 | 59 | 93 | 0.871 | 0.388 | 0.686 | 0.656 | 0.483 |
| TLSQ_20 | 0.20 | 26 | 5 | 45 | 107 | 0.839 | 0.296 | 0.634 | 0.727 | 0.543 |
| TLSQ_45 | 0.45 | 18 | 13 | 14 | 138 | 0.581 | 0.092 | 0.438 | 0.852 | 0.489 |
| TLSQ_50 | 0.50 | 16 | 15 | 13 | 139 | 0.516 | 0.086 | 0.448 | 0.847 | 0.431 |
| TLSQ_55 | 0.55 | 13 | 18 | 11 | 141 | 0.419 | 0.072 | 0.458 | 0.842 | 0.347 |
| TLSQ_60 | 0.60 | 10 | 21 | 9 | 143 | 0.323 | 0.059 | 0.474 | 0.836 | 0.263 |
| TLSQ_65 | 0.65 | 9 | 22 | 9 | 143 | 0.290 | 0.059 | 0.500 | 0.831 | 0.231 |
| TLSQ_90 | 0.90 | 4 | 27 | 2 | 150 | 0.129 | 0.013 | 0.333 | 0.842 | 0.116 |

**Table 14.** Summary of contingencies and efficiency values (Ef) of the forecasts of SLEs and NSLEs using the three best thresholds computed on the 2008–2016 dataset before (**A**) and after (**B**) the reduction of the dataset shown in Figures 8 and 9.

|   | Threshold | $p$, $p_e$ | TP | FN | FP | TN | Ef |
|---|---|---|---|---|---|---|---|
| **A** | LR_50 | 0.50 | 13 | 18 | 3 | 533 | 0.963 |
|   | LSQ_45 | 0.45 | 18 | 13 | 4 | 522 | 0.952 |
|   | QR_50 | 0.50 | 15 | 16 | 13 | 523 | 0.949 |
| **B** | LR_50 | 0.50 | 13 | 18 | 3 | 149 | 0.885 |
|   | LSQ_45 | 0.45 | 18 | 13 | 4 | 138 | 0.852 |
|   | QR_50 | 0.50 | 15 | 16 | 13 | 139 | 0.842 |

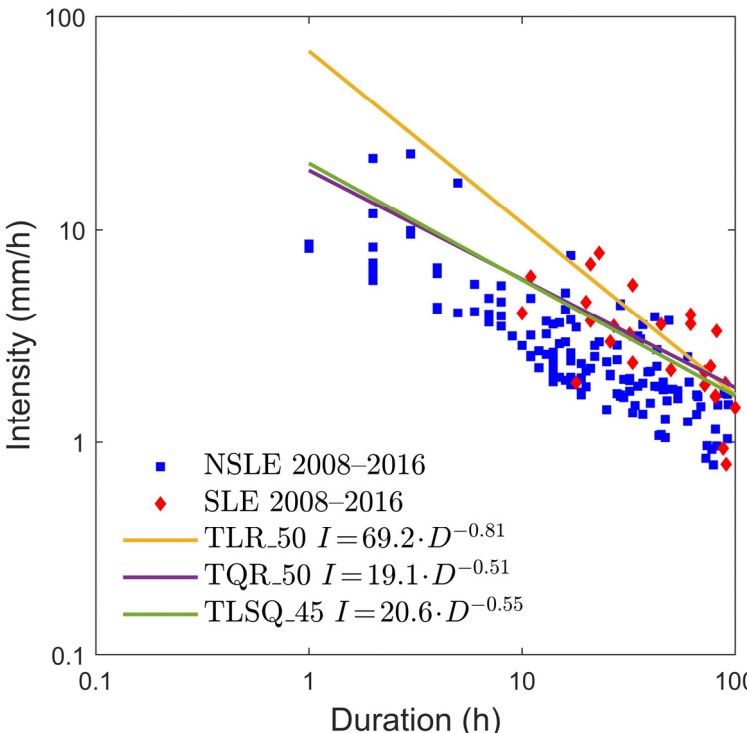

**Figure 14.** Rainfall thresholds with maximum efficiency computed for the 2008–2016 period shown in Table 14B obtained with the three different statistical methods. In this case there is a strong similarity between the QR and LSQ thresholds, reminiscent of the 1975–2002 period. Blue squares: events that did not triggered landslides (NSLE); red lozenges: events that triggered landslides (SLE).

The rainfall thresholds calculated by applying the three statistical methods on the dataset of 183 events produced results (Table 14B) which, in terms of Ef, are between 0.84 and 0.88, with 162 out of 183 events correctly predicted. This proves to be comparable to the results of the thresholds obtained by updating the work of [27] and shown in Table 4.

*4.3. Combination of 2008–2016 SLE and NSLE Events and the Thresholds Calculated on the Dataset Related to the 1975–2002 Period*

After the calculation of the rainfall threshold curves through the three fit methodologies (LR, QR, LSQ) on the respective datasets (1975–2002 and 2008–2016 periods), the last statistical processing carried out in this work aimed to validate the thresholds calculated regarding the 1975–2002 period by joining them with a different dataset (in this case, the most recent data of the 2008–2016 period). Using the R software, the 2008–2016 dataset was combined with the parameters of the threshold curves calculated on the 1975–2002 dataset, obtaining a new series of three graphs (Figures 15–17) and contingency tables (Tables 15–17).

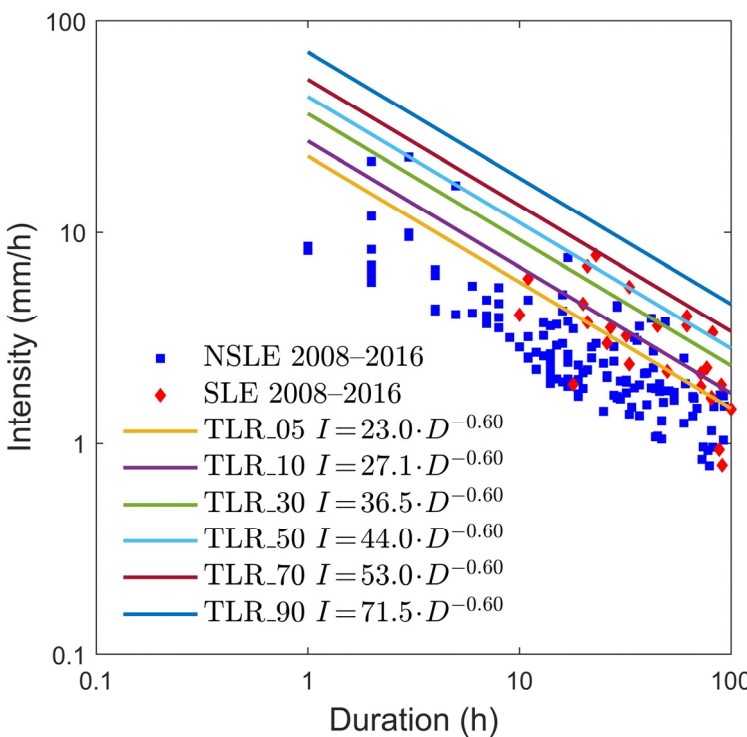

**Figure 15.** Rainfall thresholds obtained using the LR technique. Blue squares: events that did not trigger landslides (NSLE); red lozenges: events that triggered landslides (SLE).

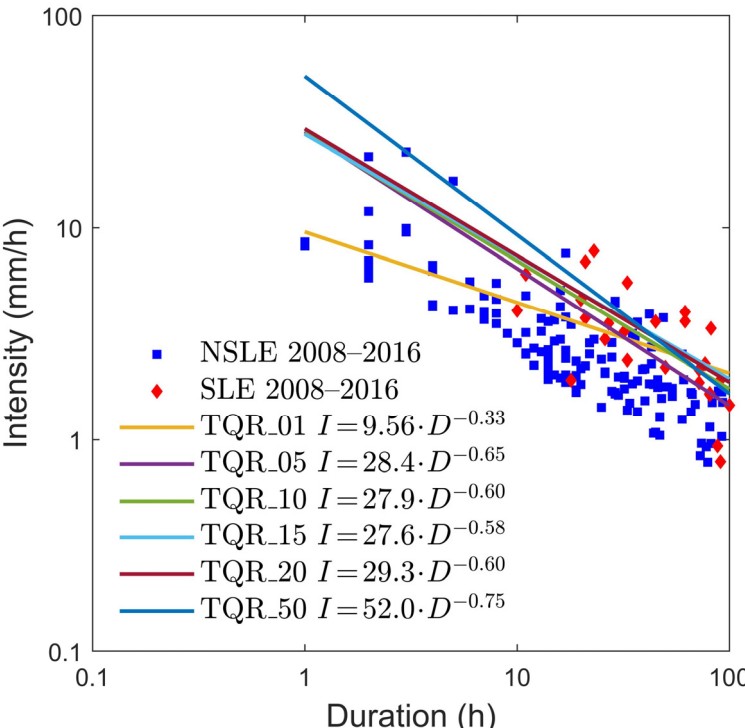

**Figure 16.** Rainfall thresholds obtained using the QR technique. Blue squares: events that did not trigger landslides (NSLE); red lozenges: events that triggered landslides (SLE).

As in the previous paragraphs, in this new comparison between the obtained rainfall thresholds, the three best thresholds in terms of efficiency and accuracy in forecasting SLE and NSLE events were highlighted in Table 18 and in Figure 18.

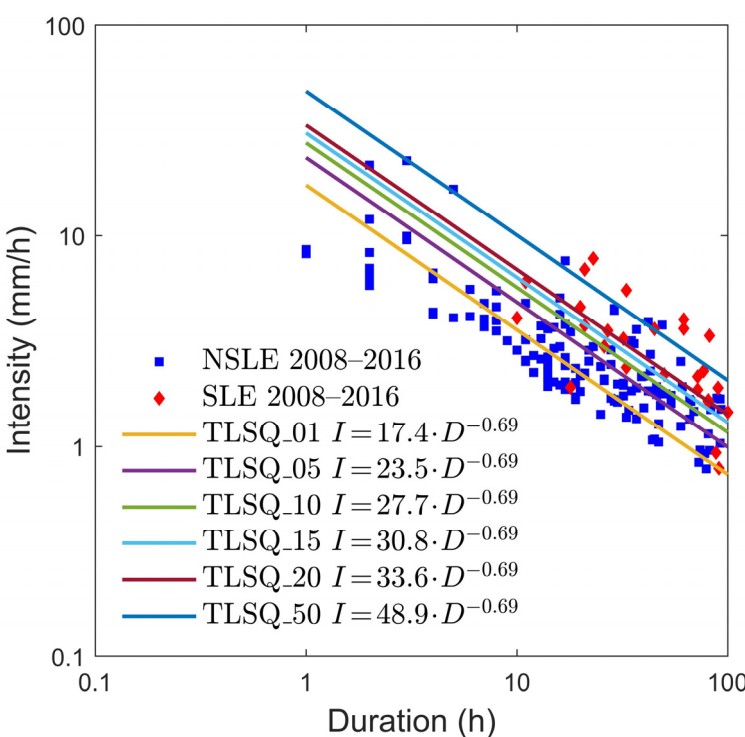

**Figure 17.** Rainfall thresholds obtained using the LSQ technique. Blue squares: events that did not triggered landslides (NSLE); red lozenges: events that triggered landslides (SLE).

**Table 15.** Contingency table (TP, FN, FP, TN) and skill scores (POD, POFD, POFA Ef, HK) for the dataset of the 2008–2016 period and the rainfall thresholds calculated regarding the 1975–2002 dataset obtained using the LR method. Maximum efficiency value is reached with the curve highlighted in red relating to a landslide probability of 0.1 (Ef = 0.858). NA: not available.

| Threshold | p | TP | FN | FP | TN | POD | POFD | POFA | Ef | HK |
|-----------|------|----|----|----|-----|-------|-------|-------|-------|-------|
| TLR_5 | 0.05 | 20 | 11 | 17 | 135 | 0.645 | 0.112 | 0.459 | 0.847 | 0.533 |
| TLR_10 | 0.10 | 15 | 16 | 10 | 142 | 0.484 | 0.066 | 0.400 | 0.858 | 0.418 |
| TLR_30 | 0.30 | 7 | 24 | 4 | 148 | 0.226 | 0.026 | 0.364 | 0.847 | 0.199 |
| TLR_50 | 0.50 | 5 | 26 | 0 | 152 | 0.161 | 0.000 | 0.000 | 0.858 | 0.161 |
| TLR_70 | 0.70 | 0 | 31 | 0 | 152 | 0.000 | 0.000 | NA | 0.831 | 0.000 |
| TLR_90 | 0.90 | 0 | 31 | 0 | 152 | 0.000 | 0.000 | NA | 0.831 | 0.000 |

**Table 16.** Contingency table (TP, FN, FP, TN) and skill scores (POD, POFD, POFA Ef, HK) for the 2008–2016 dataset and the thresholds calculated regarding the data relating to the 1975–2002 period obtained using the QR method. Maximum efficiency value is reached with the curve highlighted in red relating to an exceedance probability of 0.5 (Ef = 0.863), which is considered better for the higher POD and HK parameters.

| Threshold | $p_e$ | TP | FN | FP | TN | POD | POFD | POFA | Ef | HK |
|-----------|------|----|----|----|-----|-------|-------|-------|-------|-------|
| TLR_01 | 0.01 | 14 | 17 | 26 | 126 | 0.452 | 0.171 | 0.650 | 0.765 | 0.281 |
| TLR_05 | 0.05 | 19 | 12 | 18 | 134 | 0.613 | 0.178 | 0.486 | 0.836 | 0.494 |
| TLR_10 | 0.10 | 14 | 17 | 10 | 142 | 0.452 | 0.066 | 0.417 | 0.852 | 0.386 |
| TLR_15 | 0.15 | 11 | 20 | 9 | 143 | 0.355 | 0.059 | 0.450 | 0.842 | 0.296 |
| TLR_20 | 0.20 | 12 | 19 | 9 | 143 | 0.387 | 0.059 | 0.429 | 0.847 | 0.328 |
| TLR_50 | 0.50 | 14 | 17 | 8 | 144 | 0.452 | 0.053 | 0.364 | 0.863 | 0.399 |
| TLR_55 | 0.55 | 7 | 24 | 1 | 151 | 0.226 | 0.007 | 0.125 | 0.863 | 0.219 |
| TLR_60 | 0.60 | 7 | 24 | 1 | 151 | 0.226 | 0.007 | 0.125 | 0.863 | 0.219 |
| TLR_65 | 0.65 | 7 | 24 | 0 | 152 | 0.226 | 0.000 | 0.000 | 0.869 | 0.226 |
| TLR_90 | 0.90 | 5 | 26 | 0 | 152 | 0.161 | 0.000 | 0.000 | 0.858 | 0.161 |

**Table 17.** Contingency table (TP, FN, FP, TN) and skill scores (POD, POFD, POFA, Ef, HK) for the 2008–2016 dataset period and the thresholds calculated on the data relating to the 1975–2002 period obtained using the LSQ method. Maximum efficiency value is reached with the curve highlighted in red relating to an exceedance probability of 0.30 (Ef = 0.858). NA: not available.

| Threshold | $p_e$ | TP | FN | FP | TN | POD | POFD | POFA | Ef | HK |
|---|---|---|---|---|---|---|---|---|---|---|
| TLR_01 | 0.01 | 30 | 1 | 92 | 60 | 0.968 | 0.605 | 0.754 | 0.492 | 0.362 |
| TLR_05 | 0.05 | 26 | 5 | 51 | 101 | 0.839 | 0.336 | 0.662 | 0.694 | 0.503 |
| TLR_10 | 0.10 | 25 | 6 | 31 | 121 | 0.806 | 0.204 | 0.554 | 0.798 | 0.603 |
| TLR_15 | 0.15 | 23 | 8 | 24 | 128 | 0.742 | 0.158 | 0.511 | 0.825 | 0.584 |
| TLR_20 | 0.20 | 20 | 11 | 22 | 130 | 0.645 | 0.145 | 0.524 | 0.820 | 0.500 |
| TLR_30 | 0.30 | 14 | 17 | 9 | 143 | 0.452 | 0.059 | 0.391 | 0.858 | 0.392 |
| TLR_50 | 0.50 | 9 | 22 | 4 | 148 | 0.290 | 0.026 | 0.308 | 0.858 | 0.264 |
| TLR_55 | 0.55 | 8 | 23 | 2 | 150 | 0.258 | 0.013 | 0.200 | 0.863 | 0.245 |
| TLR_60 | 0.60 | 7 | 24 | 0 | 152 | 0.226 | 0.000 | 0.000 | 0.869 | 0.226 |
| TLR_65 | 0.65 | 6 | 25 | 0 | 152 | 0.194 | 0.000 | 0.000 | 0.863 | 0.194 |
| TLR_90 | 0.90 | 0 | 31 | 0 | 152 | 0.000 | 0.000 | NA | 0.831 | 0.000 |

**Table 18.** Summary of the contingencies and the skill scores of the forecasts of the SLEs and NSLEs using the three best thresholds computed with the threshold parameters of the 1975–2002 dataset combined with the 2008–2016 dataset.

| Threshold | $p, p_e$ | TP | FN | FP | TN | POD | POFD | POFA | Ef | HK |
|---|---|---|---|---|---|---|---|---|---|---|
| TLR_10 | 0.10 | 15 | 16 | 10 | 142 | 0.484 | 0.066 | 0.400 | 0.858 | 0.418 |
| TQR_50 | 0.50 | 14 | 17 | 8 | 144 | 0.452 | 0.053 | 0.364 | 0.863 | 0.399 |
| TLSQ_30 | 0.30 | 14 | 17 | 9 | 143 | 0.452 | 0.059 | 0.391 | 0.858 | 0.392 |

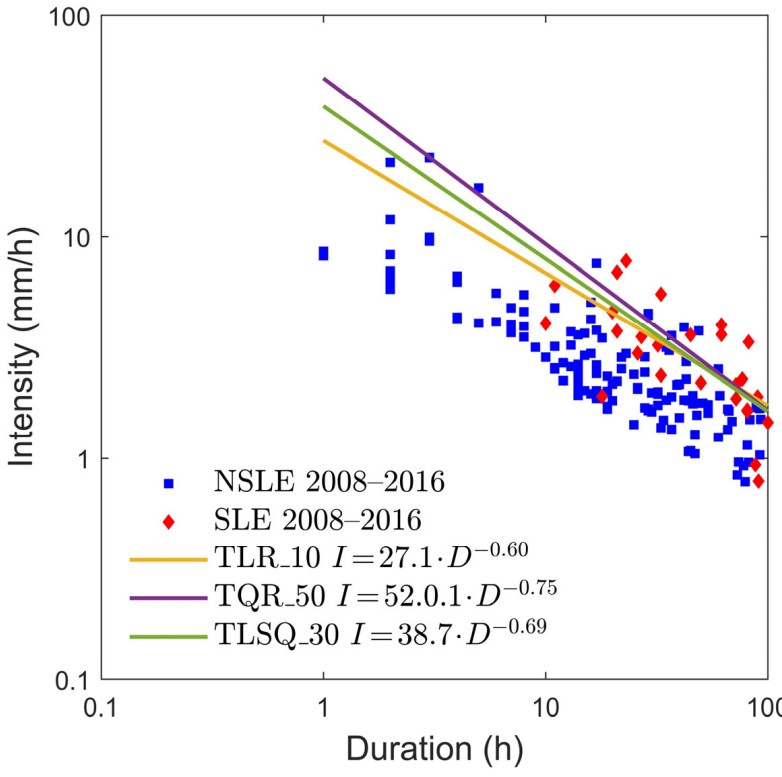

**Figure 18.** The three rainfall thresholds with maximum efficiency computed using the combination of the rainfall data of the 2008–2016 period and the threshold parameters of the 1975–2002 period. Blue squares: events that did not triggered landslides (NSLE); red lozenges: events that triggered landslides (SLE).

The validation results made by combining the rainfall thresholds of the 1975–2002 period with the dataset of 183 rainfall events relating to the 2008–2016 period highlighted in Table 18 and Figure 18 are comparable to the results obtained previously for the rainfall thresholds calculated on their respective input data shown in Tables 7 and 14B. The thresholds of the 1975–2002 period applied on the NSLE and SLE events of the 2008–2016 have Ef values between 0.85 and 0.86, with a maximum of 158 correctly predicted events, and a QR threshold at the fiftieth percentile of exceedance probability.

## 5. Discussion

In order to obtain the rainfall thresholds for shallow landslide occurrence in an area prone to landslides (like the southern Apuan Alps), in this study we used three different statistical procedures (LR, QR, LSQ). We compared the results and then further compared them with old threshold curves obtained by [27] using non-statistical and repeatable approaches. Two different datasets were analyzed: a first rainfall event dataset concerning the 1975–2002 period (the same used by [27]) and one concerning the 2008–2016 period. In both datasets, the rainfall events were classified into two categories: events inducing shallow landslides (SLE) and events not inducing shallow landslides (NSLE). LSQ and QR techniques allowed us the ability to assess the ID rainfall thresholds for several exceeding rainfall probabilities, and we specifically show the results for exceeding probabilities of 1%, 5%, 10%, 15%, 20%, and 50%). The ID thresholds for the range of landslide occurrence probability from 5% to 90% were individuated using the LR method on the same datasets, and we specifically show the landslide probability results of 5%, 10%, 30%, 70%, 90%).

The results obtained using both analyzed datasets highlight the validity of the three statistical techniques used to define the thresholds. We recognized some small differences between the three statistical methods applied, both within the two different datasets (1975–2002 and 2008–2016) and between the two datasets themselves. The most significant difference has to do with the LR method. The rainfall thresholds show different slopes for the 1975–2002 dataset ($\beta = -0.60$) and 2008–2016 dataset ($\beta = -0.82$). This difference could be attributed to the different techniques applied to reconstruct the rainfall events. The hypothesis that climatic conditions may have changed can be ruled out by the fact that there does not appear to be a significant difference between the thresholds obtained with the other two methods. However, it is interesting to explore this issue [61,62] since climatic conditions are changing, including rainfall and extreme events [63], and constant verification and updating of thresholds will be necessary.

The comparison between old rainfall thresholds of [27], obtained using empirical techniques, and new ones, obtained by LR, QR, and LSQ statistical approaches using the same dataset, highlight how the latter are always included between the upper and lower thresholds of [27], but the differences seem to not be particularly relevant.

Regardless of the methodology used, when elaborating rainfall thresholds inducing shallow landslides in a given area, some aspects are essential. The first one is the exact knowledge of the characteristics of the rainfall event (start and end time, duration, and cumulated rainfall). This depends on the availability of a dense monitoring network that can cover the study area and is equipped with efficient automatic instruments that are able to record the rainfall data continuously. If this is not possible (e.g., raingauge far from the area, imperfect functioning of the instruments, etc.), the results are probably affected by errors. A second fundamental aspect is the information about the effects of the rainfall, obviously in terms of landslides activated. For recent events, specific field surveys could be carried out, however for past events this information is often lacking and may be obtained by indirect sources such as archive research. On the other hand, for statistical analysis it is important to cover large periods of time with many events in order to obtain stronger and more reliable relationships to separate NSLEs and SLEs based on the characteristics of the rainfall. Archive research may not be efficient. For example, we can obtain vague or inaccurate information on landslide time or location, or even find no evidence of landslides despite the fact that they did occur (simply because they did not provide any news, having

not interfered with human activity or infrastructures). The lack of accuracy of rainfall datasets and landslides is the main limitation for this type of study.

However, there is another important constraint, namely the characteristics of the area, from different points of view: geological, geomorphological, land use, climate, and so on. Shallow landslides can be triggered in many landscapes, but in different ways depending on the conditions mentioned. Therefore, the choice of a homogeneous area such as the southern Apuan Alps is fundamental for having a reliable database.

## 6. Conclusions

Many methods for the rainfall thresholds individuation are used worldwide to determine the rainfall characteristics (in terms of duration, intensity, cumulated rainfall, and other similar parameters) which can initiate shallow landslides. Nevertheless, the great experience and numerous attempts of the many researchers who have dedicated themselves to this type of research cannot avoid the fact that different approaches can produce results that are not always comparable. Furthermore, the reliability of the databases used significantly affects the results obtained. Despite these limitations, we think that these methodologies can offer a valid support, obtainable with reasonable costs and time, which can help authorities to manage alert systems in shallow landslide-prone areas, especially in small and homogeneous environments. Indeed, it is crucial to underline again that choosing a small and homogeneous area allows us to obtain thresholds high enough not to issue a state of alert, even in the case of a not-particularly-intense meteorological phenomena. In this way, it is possible to avoid having a high number of false alarms, as sometimes happens in the case of thresholds calculated on a larger scale (regional or even national).

Specifically, the southern Apuan Alps include these characteristics: a small homogeneous area and high susceptibility to heavy rainstorms inducing shallow landslides. These conditions are optimal for the application of the rainfall threshold's methods in their best conditions, namely achieving a limited number of false and missed alarms. In this research, three statistical approaches (logistic regression (LR), quantile regression (QR), and least-squares linear fit (LSQ)), were applied for the first time in the study area to obtain the rainfall thresholds able to initiate shallow landslides, also comparing them with a previous attempt made by [27] using a non-statistical technique. These approaches were applied to two different datasets, including rainfall data and landslide occurrence information, related to the periods of 1975–2002 and 2008–2016, respectively. The best thresholds obtained for each dataset in terms of efficiency are TLR_55 (Ef = 0.898), TQR_55 (0.886), TLSQ_55 (0.892) for the 1975–2002 dataset and TLR_50 (0.885), TQR_50 (0.842), TLSQ_45 (0.852) for the 2008–2016 dataset, considering the predictive capacity of the thresholds obtained using different methodologies to be acceptable. The comparison between the rainfall thresholds computed by a simple manual fitting and the new statistical ones basing the same dataset do not show significant differences, making it possible to test the thresholds obtained for the management of alerts.

The following step Is necessarily a testing period of the obtained rainfall thresholds to verify their effectiveness and applicability, and then adjust them accordingly, before a final use of the authorities responsible for risk management in a densely populated area. Naturally, the concrete use of these thresholds by civil protection requires close coordination with the public bodies that issue weather forecasts, which should be able to provide reliable data in time for their processing.

Finally, we have been informed that a municipal administration of the study area is acquiring data on rainfall thresholds to start an experimental phase. We believe that this could be a flattering result for scientific research on this topic.

**Author Contributions:** Conceptualization, R.G. and M.B.; methodology, R.G., M.B. and A.Z.; software, R.G., M.B. and A.Z.; validation, R.G. and M.B.; formal analysis, R.G., M.B. and A.Z.; investigation, R.G., M.B. and A.Z.; resources, R.G. and M.B.; data curation, R.G. and M.B.; writing—original draft preparation, R.G., M.B. and A.Z.; writing—review and editing, R.G. and M.B.; visualization, R.G. and M.B.; supervision, R.G. and M.B. All authors have read and agreed to the published version of the manuscript.

**Funding:** This research received no external funding.

**Data Availability Statement:** The rainfall data are available at the website of the Regional Hydrological Service of Tuscany (https://www.sir.toscana.it/ (accessed on 24 May 2018). The landslide information is collected by newspapers, cited scientific papers, local authorities, etc. Requests to access the dataset should be addressed to the corresponding author.

**Acknowledgments:** The author are grateful to the reviewers, whose suggestions and comments significantly improved this manuscript.

**Conflicts of Interest:** The authors declare no conflicts of interest.

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
