# Peer review of "An Update on Rainfall Thresholds for Rainfall-Induced Landslides in the Southern Apuan Alps (Tuscany, Italy) Using Different Statistical Methods"

_water, doi:10.3390/w16050624_

Round 1

Reviewer 1 Report

Comments and Suggestions for Authors

This paper uses three statistical methods: logistic regression (LR), quantile regression (QR), and least squares linear fitting (LSQ) to analyze the I-D threshold line for landslides in the southern Alps of Italy. By analyzing various scenarios of different excess rainfall probabilities, the authors compare and analyze the advantages and disadvantages of the three methods in defining rainfall thresholds. The authors have conducted a lot of comparative analysis and obtained certain results, which are of certain reference significance to the relevant research field. However, there is already a lot of related research on the construction of I-D thresholds for rainfall-induced landslides, and readers may be more interested in threshold studies with breakthrough significance. The authors did not clearly express the improvements of this research in terms of methods or theories compared to previous relevant research. Therefore, it is suggested that the authors focus on analyzing the areas where this research has made progress, including the following aspects:

First, in terms of the selection of statistical methods, why did the authors choose logistic regression (LR), quantile regression (QR), and least squares linear fitting (LSQ), and where are the improvements of this study compared to previous studies that used these methods? The authors may need to supplement the current research status and add relevant discussions.

The authors mentioned that current relevant research calculates I-D thresholds using different methods and increasingly complex statistical techniques, but they did not summarize the specific mature methods currently available or the remaining problems, and where the improvements of this study are. Therefore, it is recommended that the authors supplement the current research status and summary, and add relevant experiments or discussions.

The purpose of this study is to compare the advantages and disadvantages of three different statistical methods in constructing rainfall thresholds, but in the conclusion, it is not very clear which method has a clear advantage and has improved the efficiency of identifying landslide events compared to other methods.

In addition, the legend in the figures of the I-D threshold line in the paper generally occupies a large proportion, while the size of the actual graph is too small. It is recommended that the author make adjustments.

Author Response

REVIEWER 1 (R1)

R1: First, in terms of the selection of statistical methods, why did the authors choose logistic regression (LR), quantile regression (QR), and least squares linear fitting (LSQ), and where are the improvements of this study compared to previous studies that used these methods? The authors may need to supplement the current research status and add relevant discussions.

Authors: The statistical methods that were used in this paper are those commonly used in many other works, both by us and by other authors. They are not the only ones but are the most widespread. These methods are considered quite standard and well-established, so they cannot be "improved". However, they provide results that are interesting for investigating the landslide susceptibility of the area under investigation. The detailed description of these methods is provided in many works cited in the bibliography, written by us or by other authors. We have tried to provide some better description of some features of the methods.

R1: The authors mentioned that current relevant research calculates I-D thresholds using different methods and increasingly complex statistical techniques, but they did not summarize the specific mature methods currently available or the remaining problems, and where the improvements of this study are. Therefore, it is recommended that the authors supplement the current research status and summary and add relevant experiments or discussions.

Authors: This study does not attempt to identify new methods for calculating the I-D thresholds but intends to use already existing (and widely used) methods without having any intention of improving them (because it would not be possible). However, by using this working methodology we believe we can provide results that are comparable with other results of the same type in other geologically similar areas. We quickly mentioned other methods, present in the literature, which we deemed appropriate not to use.

R1: The purpose of this study is to compare the advantages and disadvantages of three different statistical methods in constructing rainfall thresholds, but in the conclusion, it is not very clear which method has a clear advantage and has improved the efficiency of identifying landslide events compared to other methods.

Authors: This work is not intended to compare different methods to try to derive the best one. Indeed, the three methods used provide thresholds with very similar efficiency; if this parameter is chosen as the criterion for determining the optimal threshold, there is no method that prevails over the other two. From a statistical point of view, however, the fact that different methods provide results that are reasonably in agreement with each other makes the results obtained more robust.

R1: In addition, the legend in the figures of the I-D threshold line in the paper generally occupies a large proportion, while the size of the actual graph is too small. It is recommended that the author make adjustments.

Authors: We reduced the size of the fonts, however maintaining good readability.  We prepared figures with legends having a smaller font. We leave it up to the editor to decide which ones to use, based on the final size of the figures (which is not up to us to decide).

Reviewer 2 Report

Comments and Suggestions for Authors

In the submitted manuscript, the authors utilized different statistical techniques to identify the rainfall threshold values concerning the shallow landslide events. Two different datasets covering different periods (i.e., 1975-2002 and 2008-2016) are examined in this regard. The study area is chosen as Southern Apuan Alps, Italy, as the corresponding region have experienced several serious events. My comments regarding the submitted manuscript are as follows:

Please explain the reason for focusing on a dataset covering the years between 2008 and 2016. Section 3.1 encompasses the information regarding the utilized datasets; however, the reason for the corresponding particular selections is missing. 

Despite numerous attempts made using the statistical techniques in the pertinent literature, the Introduction lacks an adequate description of them. 

The authors are suggested to present the major novelty of this research and its distinctive features compared to those of previously published.

Please provide a legend for Figure 1.

Consistency in the citations is necessary (e.g., Line 174). You can check the journal citation format.

The authors are suggested to provide more information regarding the background of the utilized methods to identify the rainfall thresholds. Also, the manuscript lacks an adequate explanation of the datasets. Descriptive statistics can be shared and the map of the past landslide locations can be provided. 

Discussion of the obtained findings is missing. Also, please discuss how this study contributes to practical applications in dealing with landslide events. 

The authors are further suggested to provide the limitations of this research and the corresponding recommendations for future attempts.

Author Response

REVIEWER 2 (R2)

(R2) Please explain the reason for focusing on a dataset covering the years between 2008 and 2016. Section 3.1 encompasses the information regarding the utilized datasets; however, the reason for the corresponding particular selections is missing.

Authors: we improved the reason why the second dataset involved the 2008-2016 period.

(R2) Despite numerous attempts made using the statistical techniques in the pertinent literature, the Introduction lacks an adequate description of them.

Authors: We improved the description in the Materials and Methods section. The methodologies used are described in detail in many works published by us or by other authors, cited also in this manuscript.

(R2) The authors are suggested to present the major novelty of this research and its distinctive features compared to those of previously published.

Authors: The major novelty is the updating of the old manual rainfall thresholds with modern statistical techniques and their validation using a more recent dataset in the same area. An improvement was reported in the Settings, criticisms, and goals section.

(R2) Please provide a legend for Figure 1.

Authors: a description of the main symbols of Figure 1 was reported in the caption.

(R2) Consistency in the citations is necessary (e.g., Line 174). You can check the journal citation format.

Authors: The format was updated.

(R2) The authors are suggested to provide more information regarding the background of the utilized methods to identify the rainfall thresholds. Also, the manuscript lacks an adequate explanation of the datasets. Descriptive statistics can be shared and the map of the past landslide locations can be provided.

Authors: we implemented the information collected to build the dataset. The landslides information was obtained by consulting newspapers, scientific papers, technical reports, etc. They do not usually include information on the landslides sites, therefore a map is not at present available. However, we are willing to share our data with scientific groups which can contact us by mail.

(R2) Discussion of the obtained findings is missing. Also, please discuss how this study contributes to practical applications in dealing with landslide events.

Authors: in the Conclusions section we highlighted the importance of the recalculation of the old thresholds obtained with manual methods and its validation using a new dataset.

(R2) The authors are further suggested to provide the limitations of this research and the corresponding recommendations for future attempts.

Authors: we improved this part in Conclusions section.

Reviewer 3 Report

Comments and Suggestions for Authors

The authors explore the impact of rainfall on the initiation of shallow landslides in Italy. They used two data sets of landslides ranging from 1975 to 2016. They isolated landslides due to rainfall from landslides generated by other drivers, and tried to establish a relationships between rainfall events and landslide.  They applied three different statistical techniques to define the rainfall threshold that could initiate shallow landslides. Unfortunately, as written in the conclusion, the tested methods did not improve previous statistical methods.

The dataset used is pertinent because it includes a large amount of data allowing a statistical analysis and pertinent conclusions.

I have some comments and questions that allow me proposing major revisions.

General comments:

• As explained in the introduction, the initiation of landslides is driven by several processes. The authors focused their study on the rainfall-driven landslides: how are they sure that they were initiated by rainfalls? A more detailed explanation of this selection has to be added to be more confident of their conclusion.

• The analysis were done seasonally, i.e. for 4 periods of 3 months, why? Does this cut have a meaning in terms of processes? The time cut means nothing because the range is too wide to establish relationships between rainfall. I strongly suggest to process the data every week to go further in the analysis and interpretation. The fine period cut will limit the effect of transition between wide period.

• The analysis is based of rainfall events that is never really defined. The text mention that a precipitation of 0.01 mm/h as an event! This is a brine and not a rainfall!

• A map with NSLE and SLE landslides should be imperatively added to better analyze the landslide distribution.

• The figures 7, 8 and 9 very well highlight the there is no simple correlation between rainfall events and landslide initiation, and so make useless a statistical analysis trying to correlate rainfall and shallow landslide

For all these reasons, I recommend a major revision that could help the authors to largely improve the manuscript by pointing out why it is important to publish it.

Detail comments (see also PDF file with comments)

Lines 41-43 : please define « superficial landslides » and « large landslides ». Add some references that links superficial landslides with rainfall.

Line 44-45 : this sentence seems controversy to the first sentence of this section. Rewrite it or all this paragraph.

Line 50: define “surface landslides”.

Lines 81,82: provide values of permeability of these rocks, or references with values.

Line 84: provide slope values or better a slope map.

Lines 105: provide values of water content in soil.

Lines 137-150 the landslides datasets. What is an event?  How is it defined? Apparently they include several landslides. The authors should provide a map with the landslide datasets. I don’t understand the selection process to distinguish NSLE from SLE. Explain longer. What is the uncertainty? Was there an independent check of these datasets?

Line 305: unclear sentence.

Line 334-337: How are you sure that the landslides are triggered by the rainfall events? The authors made a very good remark.

Line 363: 0.01 m/h cannot be considered as a rainfall event!

Author Response

REVIEWER 3 (R3)

General comments:

(R3) As explained in the introduction, the initiation of landslides is driven by several processes. The authors focused their study on the rainfall-driven landslides: how are they sure that they were initiated by rainfalls? A more detailed explanation of this selection has to be added to be more confident of their conclusion.

Authors: as all types of landslides, even the shallow landslides could have many preparatory and triggering factors (rainfall, geotechnical, hydrogeologic, geological and morphological features, soil humidity, land use, human activity, etc.). Right for these reasons, it is rare, if not impossible, to have a lot of data to analyze the stability conditions of slopes over a large area (e.g., using classic deterministic approaches with the factor of safety, etc.). For superficial landslides, many studies (some of which are also reported in the bibliography) have a closer correlation with rainfall, especially intense rainfall, and usually rainfall is an information often available. The answer to the comment is therefore that in fact we can never be sure that the only factor is the rainfall, but having a lot of information on rainfall and landslides allows us to try to identify the values ​​of the rain (e.g. intensity and duration) which increases the risk of landslides occurring. However, we tried to implement this in the manuscript too.

(R3) The analysis was done seasonally, i.e. for 4 periods of 3months, why? Does this cut have a meaning in terms of processes? The time cut means nothing because the range is too wide to establish relationships between rainfall. I strongly suggest processing the data every week to go further in the analysis and interpretation. The fine period cut will limit the effect of transition between wide periods.

Authors: The seasonally analysis is done only to determine which are the seasons in which the intense rainfall events are more frequent. This analysis was previously made by Giannecchini (2006) on the events triggered over the 1975-2002 period and replicated here over the 2008-2016 period to compare the results.

(R3) The analysis is based of rainfall events that is never really defined. The text mention that a precipitation of 0.01 mm/h as an event! This is a brine and not a rainfall!

Authors: in the Materials and Methods section the characteristics of the rainfall events considered are described. A precipitation of 0.01 mm/h is not obviously an event, but in this type of analysis we must always consider the intensity together with the duration: an event of 1 mm/h is nothing if it lasts 1 hour, but if lasts 100 hours it could induce effects.

(R3) A map with NSLE and SLE landslides should be imperatively added to better analyze the landslide distribution.

Authors: as explained in the Materials and Methods section, information on landslides were collected by archive research (newspapers, scientific papers, authorities report, etc.), not by a field survey, therefore a landslides map is not available. NSLE are the rainfall events that did not initiate landslides.

(R3) The figures 7, 8 and 9 very well highlight the there is no simple correlation between rainfall events and landslide initiation, and so make useless a statistical analysis trying to correlate rainfall and shallow landslide

Authors: Figures 7 and 8 show a seasonal analysis of rainfall, done only to determine which are the seasons in which the intense rainfall events are more frequent. In figure 9 all the rainfall events are presented, before discarding the non-relevant ones. In none of the three pictures is presented a correlation between rainfall events and landslide initiation. The statistical analyses, shown in figures 3, 4, 5, 6, 11, 12, 13, 14, 15, 16, 17, 18, follow the methodology described in the text and must be obviously interpreted in a probabilistic sense.

Detail comments (see also PDF file with comments)

(R3) Lines 41-43: please define « superficial landslides » and « large landslides ». Add some references that links superficial landslides with rainfall.

Authors: Thank you for your suggestion: we try to better explain this difference in the text, also adding some references.

(R3) Line 44-45: this sentence seems controversy to the first sentence of this section. Rewrite it or all this paragraph.

Authors: we are sorry, but do not understand this controversy. We explained the importance to have tools to foresee shallow landslides, which are very hazardous in relation to the listed factors.

(R3) Line 50: define “surface landslides”.

Authors: we replaced the word “surface” with “shallow”, hoping it is clearer now.

(R3) Lines 81,82: provide values of permeability of these rocks, or references with values.

Authors: the hydraulic conductivity of these rocks is not available. We attributed them a relative permeability (high, low) only by general literature on the basis of their geological characteristics.

(R3) Line 84: provide slope values or better a slope map.

Authors: we add slope values.

(R3) Lines 105: provide values of water content in soil.

Authors: unfortunately, this datum, very interesting and important, is not available. It requires a structured monitoring which was never funded.

(R3) Lines 137-150 the landslides datasets. What is an event? How is it defined? Apparently they include several landslides. The authors should provide a map with the landslide datasets. I don’t understand the selection process to distinguish NSLE from SLE. Explain longer. What is the uncertainty? Was there an independent check of these datasets?

Authors: as described in the paragraph 3.1, the rainfall events considered in this work have two end members: high intensity and short duration and low intensity and high duration. Not all the rainfall events induced landslides and this finding was verified by the archive information collected. Events for which there is no information of landslides have been defined as NSLE (acronym of No Shallow Landslides Events); the reported landslide events have been defined as SLE (Shallow Landslides Events). We tried to better explain these differences in the text. The uncertainty is not evaluable with this approach, and it was not possible operate differently with these types of datasets. On the other hand, this the usual way used in these types of research to obtain rainfall thresholds.

(R3) Line 305: unclear sentence.

Authors: we tried to better explain the meaning of the sentence.

(R3) Line 334-337: How are you sure that the landslides are triggered by the rainfall events? The authors made a very good remark.

Authors: we improved the introduction section to better explain the relationship between rainfall and landslides.

(R3) Line 363: 0.01 m/h cannot be considered as a rainfall event!

Authors: Actually, in the text there was a typo: 10 mm/h for 1 hour event and 1 mm/h for 100 hours event, we improved this in the manuscript. As replied in a previous comment, the intensity must be always associated to the duration, it is not considerable separately. An event with Intensity=0.01 mm/h is not a rainfall event if Duration=1 h (we discarded them from the analysis). But an event with Intensity=0.01 mm/h and duration=100h could induced landslides. The threshold curves elaborated consider a maximum duration of 100h and a minimum intensity of almost 1 mm/h (please, see the graphs).

Round 2

Reviewer 1 Report

Comments and Suggestions for Authors

From the author's response, it can be inferred that the methods used in this study are commonly used by the author and other researchers to analyze rainfall thresholds. However, there have been no improvements or optimizations made based on existing research. This study only provides calculation results similar to those of other regions using the widely used methods. Therefore, this research does not provide an attractive method or viewpoint for establishing landslide rainfall thresholds. It simply replicates existing work in a new area. Therefore, it is suggested that the author incorporate more appealing research work to attract readers' attention to this study.

Author Response

REVIEWERS’ COMMENTS

REVIEWER 1 (R1)

Comments and suggestions:

R1: From the author's response, it can be inferred that the methods used in this study are commonly used by the author and other researchers to analyze rainfall thresholds. However, there have been no improvements or optimizations made based on existing research. This study only provides calculation results similar to those of other regions using the widely used methods. Therefore, this research does not provide an attractive method or viewpoint for establishing landslide rainfall thresholds. It simply replicates existing work in a new area. Therefore, it is suggested that the author incorporate more appealing research work to attract readers' attention to this study.

Authors: Thank you for your comment. You have perfectly understood the purpose of this research, which is not to develop new methods, but to use known statistical methods, comparing them with each other, to verify their effectiveness and comparing them with traditional manual fitting methods, applied in the past. In our opinion this area is strategic for testing these methodologies, in relation to its homogeneity from a geological, geomorphological, and climatic points of view, being small, and to high shallow landslides hazard. We tried to implement this aspect in the manuscript, in particular revising discussion and conclusions sections, also taking into account reviewer 3’ suggestions.

Reviewer 2 Report

Comments and Suggestions for Authors

Thanks for addressing my comments.

Author Response

REVIEWERS’ COMMENTS

REVIEWER 2 (R2)

Comments and suggestions:

(R2) Thanks for addressing my comments.

Authors: Thank you so much for appreciating our contributions according to your suggestions, which effectively improved the manuscript.

Reviewer 3 Report

Comments and Suggestions for Authors

The authors considered most of the comments that partially improve the manuscript. However, I am not always convinced by the interest of these statistical methods, as the authors wrote. But they can publish it, because it allows other people to use these methods.

The authors rewrote the conclusion, but no the conclusion id very very long ans is more a discussion than a conclusion. So change to conclusion into a discussion and add a short conclusion.

Author Response

REVIEWERS’ COMMENTS

REVIEWER 3 (R3)

Comments and suggestions:

(R3) The authors considered most of the comments that partially improve the manuscript. However, I am not always convinced by the interest of these statistical methods, as the authors wrote. But they can publish it, because it allows other people to use these methods.

The authors rewrote the conclusion, but no the conclusion id very very long and is more a discussion than a conclusion. So change to conclusion into a discussion and add a short conclusion.

Authors: Thank you for the suggestion about the conclusion setting. We updated the manuscript separating the discussion section and the conclusion section, implemented accordingly. We also corrected the text following the observations in the pdf.

Round 3

Reviewer 1 Report

Comments and Suggestions for Authors

Thank you for making the modifications. However, there are still some issues that need to be addressed or clarified:

1.In the abstract, it is mentioned that this method is being attempted for the first time. Could you clarify if this is the first time this method has been applied in this research field, or the first time it has been used in this study area? If this method has been previously used for rainfall threshold calculation, please modify the statement.

2.In the abstract, it is mentioned that the objective of this study is to update, improve, validate, and compare rainfall threshold calculation methods using different statistical approaches. However, it does not provide details on where the updates and improvements have been made. This is unfriendly to readers because the most relevant aspect for readers is the updates and improvements made in this study regarding rainfall threshold calculation methods. Please provide a complete description of the specific updates and improvements in the abstract.

3.In the conclusions section, the importance of attempting this method in the study area and the prospects for its applicability are discussed, but there is a lack of summarizing the results obtained in this study. Please provide a summary of the results and conclusions of this study.

Author Response

REVIEWERS’ COMMENTS

REVIEWER 1 (R1)

Comments and suggestions:

R1:

  1. In the abstract, it is mentioned that this method is being attempted for the first time. Could you clarify if this is the first time this method has been applied in this research field, or the first time it has been used in this study area? If this method has been previously used for rainfall threshold calculation, please modify the statement.
  2. In the abstract, it is mentioned that the objective of this study is to update, improve, validate, and compare rainfall threshold calculation methods using different statistical approaches. However, it does not provide details on where the updates and improvements have been made.

This is unfriendly to readers because the most relevant aspect for readers is the updates and improvements made in this study regarding rainfall threshold calculation methods. Please provide a complete description of the specific updates and improvements in the abstract.

  1. In the conclusions section, the importance of attempting this method in the study area and the prospects for its applicability are discussed, but there is a lack of summarizing the results obtained in this study. Please provide a summary of the results and conclusions of this study.

Authors:

  1. We applied the statistical methods for the first time in the study area. We better specify this issue in the abstract.
  2. We have tried to implement these aspects in the abstract. Unfortunately, the 200-word limit does not allow us to explain some concepts fully. We hope that the new version highlights the conclusions better.
  3. One of the reviewers required an improvement in the separation between the discussion and conclusion sections. In this passage, we probably lose some parts. We implemented this separation again, hoping it can be more thorough and comprehensible now.
